# Delivery Systems for Nucleic Acids and Proteins: Barriers, Cell Capture Pathways and Nanocarriers

**DOI:** 10.3390/pharmaceutics13030428

**Published:** 2021-03-22

**Authors:** Julian D. Torres-Vanegas, Juan C. Cruz, Luis H. Reyes

**Affiliations:** 1Grupo de Diseño de Productos y Procesos (GDPP), Department of Chemical and Food Engineering, Universidad de los Andes, Bogotá 111711, Colombia; 2Department of Biomedical Engineering, Universidad de los Andes, Bogotá 111711, Colombia

**Keywords:** gene therapy, internalization, nanovehicles, delivery

## Abstract

Gene therapy has been used as a potential approach to address the diagnosis and treatment of genetic diseases and inherited disorders. In this line, non-viral systems have been exploited as promising alternatives for delivering therapeutic transgenes and proteins. In this review, we explored how biological barriers are effectively overcome by non-viral systems, usually nanoparticles, to reach an efficient delivery of cargoes. Furthermore, this review contributes to the understanding of several mechanisms of cellular internalization taken by nanoparticles. Because a critical factor for nanoparticles to do this relies on the ability to escape endosomes, researchers have dedicated much effort to address this issue using different nanocarriers. Here, we present an overview of the diversity of nanovehicles explored to reach an efficient and effective delivery of both nucleic acids and proteins. Finally, we introduced recent advances in the development of successful strategies to deliver cargoes.

## 1. Introduction

Gene therapy has been considered a promising therapeutic strategy, and it is based on the delivery of genes to treat several acute acquired and inherited diseases [1]. Some examples include, among others, the autosomal or X-linked recessive single-gene disorders (i.e., cystic fibrosis), Severe Combined Immunodeficiency Defect (SCID), emphysema, retinitis pigmentosa, sickle-cell anemia, phenylketonuria, hemophilia, Duchenne Muscular Dystrophy (D.M.D), some autosomal dominant disorders, even polygenic disorders, various forms of cancers, vascular disease, neurodegenerative disorders, inflammatory conditions [2].

Viruses were the first carriers for delivering therapeutic genes, assuring protection, and taking advantage of the virus-life cycle [1]. This type of carrier, known as a viral vector, is one of the most widely used vectors in gene therapy due to its ability to carry genes efficiently and ensure long-term expression [2]. However, these vectors have several disadvantages, such as the potential risk of harmful immune responses [3], the high cost and difficulty related to their preparation [1,4], and the limited size of the genetic sequences that can be inserted into human cells [1,5]. Accordingly, there is a need to look for safer and cheaper alternatives [1]. Hence, non-viral approaches have risen to deal with the limitations of viral systems. Research in this field has attracted significant attention because of the advantages that non-viral systems offer over the viral ones regarding safety, relatively low immune response, and ease of preparation to enable large amounts at low cost [6].

Non-viral DNA delivery systems are classified into two groups: physical approaches and chemically constructed vectors. Physical approaches rely on a physical force to weaken the cell membrane, thereby facilitating the gene’s insertion into the nucleus [6]. Some strategies following this approach include electroporation, gene gun, ultrasound, and hydrodynamic delivery. Meanwhile, chemically constructed vectors can be prepared by the electrostatic interaction between polycationic derivatives, either lipids or polymers and the anionic phosphate of DNA to form a particle [6]. This complex is known as a polyplex when the interaction occurs between the polymer and DNA or a lipoplex when the DNA interacts with a phospholipid. Moreover, it is possible to chemically build DNA vectors by encapsulation or adsorption within biodegradable spherical structures to yield micro and nanoparticles [1,7]. Other chemically constructed vectors include the conjugation of bioactive compounds such as proteins and peptides in the surface of metal, magnetic, lipid, polymer, and carbon-based nanomaterials [8].

DNA plays a crucial role in storing genetic information, which is transcribed into messenger RNA. This transcript serves as the bridge between the genetic information encoded in DNA and its protein translation. Other types of nucleic acids that can potentially control protein expression include interference RNA (RNAi) and antisense oligonucleotides. These therapeutic approaches are similar as they act as a gene silencing mechanism. The ultimate goal of gene therapy is to deliver a transgene into the nucleus (for DNA delivery) or the cytoplasm (e.g., to deliver RNAi) to finally express the therapeutic protein [9]. Therefore, DNA must be complexed with the delivery system or vector that carries the therapeutic gene into the targeted cell, thereby avoiding its degradation and ensuring its final transcription [10].

Nevertheless, the nanovehicle must overcome several obstacles to enter the cell and deliver the cargo. Such barriers may be found at both the extracellular (e.g., depuration by organs like kidney, liver, and spleen, as well as the mononuclear phagocyte system (MPS)) and intracellular level (e.g., crossing the plasma membrane, encapsulation by endosomal vesicles, vector unpacking, and cytoplasmic degradation). Also, several mechanisms of internalization have been found for the cellular capture of transgenic cargoes. These may follow phagocytic routes or endocytic pathways. Based on these mechanisms, the design of nanoscale-tailored materials has been proposed as a promising alternative for the development of potential delivery systems for gene therapy. This review details both the mechanisms of delivery systems to overcome biological barriers and those of cell internalization. Moreover, this work highlights several nanomaterials that have been rationally designed as delivery systems for gene therapy.

## 2. Biological Barriers

Despite comprehensive research, non-viral vectors’ success in clinical trials has not been achieved due to their low efficiency in passing through several biological barriers [11]. These can be divided into two categories: extracellular (EC) and intracellular (IC) barriers (Figure 1).

EC barriers need to be overcome before reaching the target cell. These include organs such as the spleen, liver, and kidney. These organs are considered physiological barriers because they exhibit clearance mechanisms that uptake or filter nanoparticles out the body [13]. Other obstacles such as endo- and exonuclease activity of blood components, activation of the immune system, and finally, surpassing endothelial barriers and overcoming migration through the extracellular space [12]. Topical administration (e.g., pulmonary and oral routes) has also been intensively explored for gene therapy due to numerous advantages, such as reducing the potentially high risks of invasive methodologies (e.g., injection and surgical therapies) [14,15], improving the targeting of gene delivery systems, which is attractive for addressing more complex structures such as ocular areas [16]. Additionally, they provide more timely treatments to address disorders with a delayed symptomatic expression (e.g., laryngeal papillomatosis) [17]. In this case, the EC barriers that play an essential role are epithelial and extra factors such as the mucus and the presence of surfactants. These barriers may lead to cargo degradation or rapid clearance of nanostructures, thereby rendering them ineffective for the intended therapeutic purpose [12,18].

For a successful gene therapy approach focused on treating neurodegenerative diseases, a formidable obstacle that deserves special attention and is required to overcome is the blood-brain barrier (BBB). The BBB is present in all organisms with a well-developed central nervous system (CNS). It is composed of a neurovascular unit and a set of junctions interacting with each other, fulfilling protective functions and regulating the BBB dynamics and properties that hurdle the entry of external agents to neurons [19].

Once the EC barriers are surpassed, several IC barriers impede the free pass of vectors to achieve an efficient gene delivery. Firstly, non-viral vectors need to cross the plasma membrane to reach the IC environment [20,21]. Physical methods, such as photoporation, electroporation, or sonoporation, are sometimes used to access the cytoplasm directly [22,23,24]. However, non-viral vectors usually gain entrance to the cell by endocytosis [25,26] to localize into early endosomes that eventually mature via late endosomes into endolysosomes. This maturation is accompanied by intraluminal acidification, which reduces the endosomal pH and encompasses the activation of various degradation enzymes [25]. Hence, using endocytosis to enter the cell gives rise to an additional barrier: escaping endosomal confinement before enzymatic degradation of nucleic acids (NAs) in the endolysosomes. Also, excretion of nanovehicles from the cell via exocytosis may happen, further reducing gene delivery efficiency [27]. Even if endosomal escape happens, the NAs (whether still complexed to the vector or not) reside in the cytosol, where they must avoid clearance by autophagy or degradation via cytoplasmic nucleases [12].

Furthermore, the kinetics of cargo release from the carrier is an essential consideration since vector unpacking has been reported by Schaffer et al. [28] as another bottleneck for efficient transfection. Although some viruses demonstrated to specifically evolve and acquired mechanisms for uncoating DNA within cells, synthetic polycations may not exhibit similar efficiency for DNA release. To prove this, they developed a system composed of a cationic polymer (polylysine) linked to epidermal growth factor as the ligand to deliver pDNA encoding the green fluorescent protein to mouse fibroblasts. Co-localization studies showed that larger polylysines were found along with the nuclear plasmid. In contrast, they fail to see smaller polycations in the vicinity of such region, which might be due to differences in the affinities or dissociation rates from the plasmid within the nucleus. To prove this, polylysine-pDNA conjugates were incubated in the presence of a significant excess of dsDNA to mimic the potential dissociation of polyplexes by chromosomal DNA. This assay led them to conclude that smaller polycations dissociated from the plasmid faster than larger polycations. Afterward, this behavior was demonstrated to impact the transcriptional level significantly, as shown by in vitro studies where highly dissociating polyplexes showed higher gene expression levels than those with slower dissociation rates.

Finally, while NAs such as siRNA, mRNA, and miRNA have their site of action in the cytosol, pDNA needs to be delivered directly to the nucleus. This strongly suggests that the nuclear envelope becomes a significant barrier [12,20,21]. This was firstly corroborated in 1980 when pBR322-based plasmids were injected into the cytoplasm, and no gene expression was detected [29,30]. Other research works proved that the level of gene expression reached for plasmids injected into the cytoplasm was about only 3% of that seen when plasmids were injected directly into the nucleus [30,31]. Later studies in various mammalian cell types confirmed gene expression dependence on plasmid nuclear localization [30,32]. Therefore, the amount of DNA that may gain access to the nuclear compartment is likely to be reduced.

One way to access the nucleus is during mitosis because as this process occurs, the nuclear envelope breaks down. Therefore, plasmids in the cytoplasm can access the newly formed nuclei of daughter cells before nuclear envelope formation [30]. Fasbender et al. demonstrated that actively dividing cells were ten times more likely to express the transferred gene product [33]. Furthermore, it has been shown that the level of gene expression depends on the cell cycle. For instance, Tseng et al. [34] found an increase from 50 to 300-fold in gene expression when cells were exposed to cationic liposomes before (G2 phase) or during mitosis (G2-M phase).

However, in non-diving cells, the only way for pDNA to access the nucleus is through the nuclear pore complex (NPC), an aqueous channel in the nuclear envelope through which proteins and ribonucleoproteins can traffic during the interphase [30,35]. These pores are large multiprotein complexes composed of 100 different proteins present in multiple copies [30]. Trafficking across the nuclear envelope may occur by either signal-independent diffusion for proteins less than 50 kDa at rates inversely proportional to their size or by signal-mediated import, which needs a nuclear localization signal (NLS) moiety within the imported protein [30]. Proteins containing an NLS interact with some cytoplasmic receptor proteins named importins, which bind to both the NLS and subunits of the NPC to enable translocation across the pore. A small GTP-binding protein, called Ran, which is localized exclusively to the nucleus in its GTP-bound state, is in charge of controlling the directionality of nuclear transport and contributes to the disassembly of the NLS-importin complex on the inner face of the nuclear envelope [30].

To promote the nuclear import of pDNA into non-dividing cells, some studies have included an enhancer sequence from the DNA tumor virus S40 into the plasmid DNA to facilitate nuclear access during non-division cell cycles. This S40 enhancer, called a DNA targeting sequence (DTS), have exhibited high activity in cell lines derived from monkey, rat, mouse, hamster, chicken, and several of human origin [36]. Although the mechanism of nuclear import of the SV40 DTS remains unclear, it is known that the SV enhancer contains binding sites for a plethora of transcription factors such as AP1, AP2, AP3, NF-κB, Oct-1, TEF-I and TEF-II [37,38,39], whose primary function is to promote transcription of target genes in the nucleus. Since they are translated in the cytoplasm, NLS motifs are required within their sequences to translocate into the nucleus and bind to their corresponding DNA regulatory elements. Nonetheless, when their binding sites are presented on a plasmid in the cytoplasm, it is hypothesized that these transcription factors can bind to the exogenous DNA therein to form a DNA-protein complex thought to be responsible for moderating the interactions between DNA and importin proteins via their NLS motifs [30]. Similarly, it has been hypothesized that some promoters may interact with cell-specific transcription factors and mediate plasmid nuclear import in selected cell types such as endothelial and smooth muscle cells [40]. Other studies have demonstrated that the inclusion of NF-κB binding sites in plasmids may enhance nuclear import and subsequent gene expression [41]. Another potential DTS discovered is the *ori*P sequence from the Epstein-Barr virus (EBV) [42]. More recently, it was proven that combining the *tet* operator with a modified tetracycline repressor containing an NLS is possible to control and enhance nuclear import through DNA-protein interactions [43]. In line with this, it has been demonstrated that the use of NLS also facilitates the import of plasmids [30].

## 3. Internalization Pathways

Delivery systems must penetrate cells by breaking through the cell plasma membrane. Hence, highly regulated mechanisms with complex biomolecular interactions (e.g., interaction of nanovehicles with characteristic receptors for each of the possible cellular uptake pathways) need to occur to pass through the plasma membrane, which acts as a barrier to protect the cell’s interior from the outside environment [44]. Due to the membrane’s structural and biomolecular characteristics (i.e., a phospholipid-based bilayer membrane with proteins and other biomolecules crowded on their surface), this renders an overall negative charge with few cationic domains and selective permeability to ions, biomolecules, and nanovehicles. A deeper understanding of how these nanovehicles enter cells is vital since the underlying uptake pathways determine crucial parameters for the delivery system, including function, intracellular fate, and biological response [45,46,47]. According to the carrier’s physicochemical properties and the target cells’ lineage, internalization may be carried out by either phagocytosis or endocytic pathways.

### 3.1. Phagocytosis

Phagocytosis has a vital physiological function in protecting the organism against exogenous elements, such as infectious agents and inert particles, including drug delivery nanovehicles [48]. This mechanism occurs in various immune cells, including macrophages, neutrophils, dendritic cells, monocytes, and other non-specialized phagocytes such as fibroblasts, epithelial and endothelial cells [49,50]. The entry of external agents by phagocytosis occurs following a multistage process that includes opsonization, adhesion and ingestion, phagosome formation, and finally, phagolysosome formation (Figure 2A).

The stage of opsonization is initiated by cell surface receptors physically binding to the external nanoparticle. These receptors include Fc receptors, mannose receptors, scavenger receptors, and complemental receptors. Once a phagocyte is armed with these receptors, nanomaterials can be readily recognized and efficiently cleared from circulation [51,52]. Then, phagocytes’ recognition and clearance are mediated by the adsorption of immunoglobulins, complement proteins, and other serum proteins on the nanomaterial surface [44]. Later, nanomaterials are snared inside phagosome vesicles that finally combine with lysosomes to form phagolysosomes, which can digest foreign “*nonself*” materials (including nanomaterials) by enzymatic and biochemical reactions [44,53,54].

Blood-circulating monocytes, hepatic Kupffer cells, splenic red pulp, marginal zone macrophages, bone marrow perisinal macrophages, and liver sinusoidal endothelial cells (LSECs) are examples of MPS phagocytic cells [55]. Also, microglia are monocyte lineage cells located throughout the brain that follow phagocytic mechanisms contributing to the central nervous system’s immune defense and regulation [56]. Particularly, opsonization by complement factors, fibrinogen, and immunoglobulins, typically leads to clearance of nanomaterials from the blood by the MPS [55]. These opsonized nanomaterials are then efficiently and rapidly sequestered by macrophages and other MPS phagocytic cells. In this case, up to 99% of a systemically administered nanomaterial as a bolus dose may be sequestered by the MPS [44,55].

Opsonized nanomaterials’ highly efficient clearance poses a major challenge for the rational development of effective nanovehicles [44]. Surface functionalization of nanomaterials has been proposed to overcome their sequestration by macrophages and other phagocytic cells of the mononuclear phagocyte system [57]. For instance, gold nanoparticles of several sizes were coated with poly(ethylene glycol) (PEG) and exposed to serum proteins’ adsorption, finding smaller nanoparticles, which present a higher curved surface, give PEG more space to spread out on their surface. This leads to a reduction in the thermodynamic barrier to protein adsorption, mainly attributed to weaker PEG-PEG steric interactions in smaller PEG-coated nanoparticles. Additionally, serum protein adsorption was reduced by increasing the PEG density, which, in turn, altered the adsorbed protein layer’s composition [57]. In line with this, Dai et al. [58] found that PEG backfilling could minimize the binding of serum proteins to the surface of gold nanoparticles and, therefore, prevent these proteins from sterically blocking the binding of biorecognition molecules to specific receptor targets.

### 3.2. Endocytic Pathways

Non-phagocytic cell capture mechanisms have traditionally been referred to as pinocytosis, defined as “cell drinking” or the uptake of fluids and solutes [54]. Unlike phagocytosis, endocytic pathways occur in virtually all cells. Endocytosis may take place by four different mechanisms: clathrin-mediated endocytosis, caveolae-mediated endocytosis, macropinocytosis, and other clathrin- and caveolae-independent endocytosis [48].

#### 3.2.1. Clathrin-Mediated Endocytosis

Clathrin-mediated endocytosis (CME) mostly serves as the primary mechanism for capturing macromolecules and plasma membrane constituents [48]. Moreover, it comes about constitutively in all mammalian cells and is responsible for critical physiological roles such as nutrient uptake and intracellular communication [48,54]. The best-described type of CME mechanism is referred to as the receptor-mediated CME, in which cell membrane receptors, including transferrin receptors, low-density lipoprotein receptors, epidermal growth factor receptors, and β2 adrenergic receptors, bind to nanomaterial surface ligands and cluster them (Figure 2C) [59]. The process of clusterization involves several steps: (i) nucleation of cytosolic proteins to form a coated pit; (ii) plasma membrane bending and invagination; (iii) cutting and separation of the neck of invagination to obtain an intracellular vesicle; and (iv) uncoating of the vesicle and recovery of the endocytic proteins [44,60].

Through this mechanism, nanoparticles are entrapped in intracellular vesicles with sizes of approximately 100–500 nm [61]. Next, with the help of conformational alterations of the GTPase enzyme dynamin, vesicles are pinched off the membrane [62]. Afterward, vesicles are uncoated, allowing clathrin units (cytosolic proteins) to be recycled along with other ligands such as transferrin and riboflavin [63,64]. The resulting endocytic vesicles deliver their cargoes to “early” endosomes, which are acidified by ATP-dependent proton pumps to become late endosomes. This is followed by fusion with pre-lysosomal vesicles containing acid hydrolases to create a harsh environment that leads to the degradation of the internalized cargoes [64,65].

Another CME for non-specific adsorptive pinocytosis has been identified and referred to as fluid-phase endocytosis [64]. In this pathway, there often exist non-specific charges and hydrophobic interactions with the cell membrane that allow the entry of cargoes via clathrin-coated vesicles as described above. Internalizing receptor ligands are also placed along with these pits and extracellular fluid and its contents [48,64]. Apart from the different modes of interaction with the cell membrane, the internalization rate, in this case, is slower than receptor-dependent CME [48].

#### 3.2.2. Caveolae-Mediated Endocytosis

Caveolae-mediated endocytosis (CvME) is the second most studied and well-characterized endocytic pathway [66]. Caveolae are flask-shaped vesicles with diameters of 50–100 nm whose structure comprises proteins that are members of the caveolin protein family. These proteins have structural domains that confer scaffolding features and the possibility of binding to critical signaling molecules (Figure 2E) [66,67]. Caveolae are then formed by the co-assembly of caveolins with cytosolic coat proteins, called cavins, which are enriched in cholesterol and sphingolipids [68].

Unlike CME, CvME is a highly regulated process involving complex signaling, mainly driven by the cargo itself [64]. After particles bind to the cell surface, they move along the plasma membrane to caveolae invaginations and, through receptor-ligand interactions, are incorporated into the vesicles [48,64]. Next, the cytosolic caveolar vesicle with no enzymes present is produced by fission of the caveolae from the membrane, mediated by the GTPase dynamin [48]. Typical intracellular fate of caveolin-based vesicles includes the Golgi apparatus and the endoplasmic reticulum [69]. For this reason, CvME may provide an attractive pathway to explore the internalization of nanocarriers, mainly by applying specific nanomaterial surface engineering strategies, including the use of surface ligands such as folic acid, cholesterol, and albumin [44,48,70].

Caveolin-dependent endocytosis has also been reported to result in transcellular transport of caveolae, commonly called transcytosis, and explored in specific cell types, including endothelial, fibroblast, smooth muscle, and adipocyte cells [71,72,73]. Since endothelial cells line the blood vessels’ inner surface, these transcytosis-based pathways enable nanocarriers to penetrate them through caveolae formation and, eventually, to come across the endothelium. Therefore, the delivery of therapeutic nanovehicles to diseased tissues via transcytosis is an attractive route for superior therapeutic efficacy [44]. Particularly, receptor-mediated transcytosis has been reported as an important mechanism for transporting drugs through BBB due to receptors’ presence on the apical surface of the BBB endothelial cells [74].

#### 3.2.3. Clathrin- and Caveolin-Independent Endocytosis

Mechanisms different than those relying on clathrin and caveolin-dependent pathways have also been described comprehensively (Figure 2D). In this case, cellular entry occurs through vesicles of about 90 nm that can transport diverse cargoes. Examples include extracellular fluid, the simian virus-40 (SV40), the cholera toxin B (CTB), glycosylphosphatidylinositol (GPI)-linked proteins, interleukin-2, and several growth hormones [54]. Multiple entry pathways have been reported for these cargoes. Their internalization is mainly mediated by several effectors dependent on cholesterol and specific lipid compositions, such as Arf-6, flotillin, Cdc-42, and RhoA [75]. Most of these pathways have shown dynamin independence; however, even when this protein is involved, its role remains unclear [54]. Also, cargoes’ final fate remains unidentified, with some experimental evidence pointing towards the avoidance of Rab5-positive early endosomes. This appears to be the case of the GPI-linked proteins transported through GPI-anchored protein-enriched early endosomal compartments (GEECs) that exhibit a tubulovesicular morphology. Some of these pathways may also use GEEC-independent endosomes and interact with clathrin-dependent endocytic compartments, as it has been found, for example, in the transport of IL-2Rβ, γc cytokine receptor, and the IgE receptor FcεR1 (a major signaling pathway for allergic reactions) [54,75,76]. Another internalization subtype is independent of the GEEC and requires Arf6 positive endosomes to be recycled towards the plasma membrane. This pathway is primarily utilized by the histocompatibility class (MHC)-1 protein, critical for antigen presentation and immune response [77].

Nanomaterials modified with folate and water-soluble N-(2-hydroxypropyl) methacrylamide (HPMA), as well as virus-like particles, are some examples of non-viral vectors reported to utilize different subtypes of the clathrin- and caveolae-independent endocytosis [78]. In the case of folate-based nanostructured materials, drug targeting might be facilitated by folate binding to the GPI-anchored folate receptor, FRα, which is overexpressed in tumor cells. It has been reported that the expression of FRα increases as the cancer stage advances [54,78]. However, the folate entry into cells is complex, and along with clathrin- and caveolae-independent endocytosis, CME may be involved in specific cell types such as cisplatin-resistant human epidermoid carcinoma cells (called KB cells) and Chinese hamster ovary cells (called CHO cells) [54,79].

### 3.3. Macropinocytosis

Macropinocytosis represents a typical route for the uptake of apoptotic cell fragments [80], viruses [81], and bacteria [82]. Additionally, it contributes substantially to antigen presentation in the major histocompatibility complex II (MHCII) [69,83,84]. Through this endocytic process, cells internalize considerable volumes of extracellular fluid by large vacuoles called macropinosomes, which exhibit diameters of 0.5–10 μm (Figure 2B) [69,85]. Rather than regulation by the direct action of a receptor or cargo molecules, this mechanism is initialized by the activation of a tyrosine kinase receptor, which leads to an increase in actin polymerization, actin-mediated ruffling, and macropinosome formation [69,86]. Remarkably, some proteins such as Cdc42, Arf6, and Rab5 are found in macropinosomes and other endocytic processes, thereby indicating a relationship between macropinosome biogenesis mechanisms and different endocytic routes [69,87]. The macropinosomes exhibit a response to cytoplasmic pH and undergo acidification and fusion events and, particularly in macrophages, follow a fate like that of endosomes. Furthermore, during the stage of maturation, markers present in macropinosomes vary in composition before their fusion with lysosomes [69,88]. Particles with submicron and larger sizes may be internalized in cells that lack the phagocytoses machinery such as basophils and lymphocytes. However, in most cases, this pathway may be utilized in combination with other routes for nanomaterials’ cellular entry [54].

### 3.4. Direct Translocation

Direct translocation of the plasma membrane is a frequent internalization pathway that facilitates cationic nanoparticles’ entry into cells. This mechanism has been attributed to the electrostatic interactions between positively charged nanoparticles and negatively charged surface moieties of the plasma membrane, which are mainly developed by the presence of membrane components such as proteins, glycolipids, and phospholipids [73]. Another factor influencing the entry of nanoparticles by this pathway is particle size. It has been reported that, particles smaller than 20 nm in diameter, usually follow this pathway [89]. The strong attraction of cationic nanoparticles with the highly abundant and negatively charged components of the inner membrane layer (e.g., phosphatidylserine) generally causes the formation of transient pores that enable nanoparticle translocation towards the cytoplasm (Figure 2F) [73,89]. This behavior accounts for the efficient internalization of small cationic nanoparticles.

## 4. Nanocarriers for the Delivery of Nucleic Acids and Proteins

### 4.1. Lipid-Based Nanocarriers

Lipid-based nanocarriers include liposomes, lipid nanoparticles (LNPs), and emulsions. These are the most widely used non-viral vectors for nucleic acid (NA) therapy [90]. In the 1980s, phospholipids containing liposomes were first tested to deliver SV40 DNA to monkey kidney cells [91]. Such nanocarriers consist of spherical, self-assembled closed structures with one or several concentric lipid bilayers encircled around an inner aqueous phase (Figure 3A). The lipid coat may be composed of both cationic and ionizable lipids. In vitro and in vivo delivery of DNA, siRNA and mRNA has been enabled by liposomes synthesized from classical cationic lipids such as *N*-[1-(2, 3-dioleoyloxy)propyl]-*N*,*N*,*N*-trimethylammonium chloride (DOTMA), 1,2-dioleoyloxy-3-trimethylammonium propane chloride (DOTAP), and 1,2-dioleoyl-sn-glycero-3-phosphoethanolamine (DOPE) [90]. Recently, these liposomes demonstrated promising results in delivering mRNA to dendritic cells for cancer immunotherapy [90]. Alternatively, liposomes with ionizable lipids composed of hydrocarbon chains, linkers, and headgroups, are neutral under a physiological pH environment. However, they are ionized and protonated under the acidic conditions of endosomes and lysosomes, which further triggers osmotic lysosomes and endosome rupture [90]. As a result, this type of liposome has been successfully tested for RNAi therapies for hereditary transthyretin-mediated amyloidosis (hATTR), where they demonstrated significant endosome/lysosome escape ability [90]. Some examples of ionizable lipids previously investigated preclinically and clinically are listed in Table 1.

The resulting liposome/NA complex is obtained by electrostatic interactions between lipids (positively charged) and NAs (negatively charged). Mainly, nano-complexes derived from the interaction of liposomes and siRNAs are obtained with a slightly positive charge, thereby facilitating interaction with negatively charged cell surfaces [92]. Subsequently, these nano-complexes are delivered to cells for uptake, internalization, escape, release, and expression [90]. Despite this process’s efficiency and simplicity, significant difficulties to overcome include the cationic lipid-associated cytotoxicity, off-target effects, and limited cell types for transfection [90]. Meanwhile, recently developed advanced technologies such as bio-orthogonal liposome fusion, click chemistry, and surface engineering may be combined with lipid nanocarriers for the development of NA delivery systems that are produced more straightforwardly (i.e., with fewer manipulation steps), are more efficient, and exhibit higher precision in cell transfection [93,94,95]. These strategies look for packing and delivering NAs to cells using rapid artificial surface labeling and targeting. Instead of relying on nonspecific electrostatic interactions between the nucleic acid complex and the cell, these methods produce complexes with a bio-orthogonal functional group displayed superficially for adhesion and delivery. For example, liposomes containing ketone groups are synthesized and added to cell culture to enable ketone display on the cell surface. Then, a complementary oxyamine liposome is generated to complex with nucleic acids. Later, the oxyamine/nucleic acid lipoplex is added to the ketone from cells. Finally, oxime formation occurs at the cell surface, and the nucleic acid is endocytosed and release within the cell [93].

The ability of lipid NPs to internalize and escape endosomes has been facilitated by membrane fusion events [96] (Figure 4). In general, through hydrophobic interactions, the liposomal envelope fuses with the endosomal membrane [73]. These interactions are facilitated by the protonation of anionic groups of the liposomal envelope, thereby allowing the release of encapsulated cargoes into the cytosol [97]. Also, by incorporating cholesterol within liposomal structures, it has been possible to increase the contact sites needed for lipid mixing and pore fusion expansion [98].

**Table 1 pharmaceutics-13-00428-t001:** Some ionizable lipids employed in the production of liposomes for gene silencing.

Abbreviation	Chemical Name	Findings/Relevant Data	Reference
DLin-MC3-DMA	6*Z*,9*Z*,28*Z*,31*Z*-heptatriaconta-6,9,28,31-tetraen-19-yl-4-(dimethylamino)-butanoate	Used for the first time in Patisiran (liposome formulation).	[90]
DLin-KC2-DMA	1,2-dilinoleyl-4-(2-dimethylaminoethyl)-1,3-dioxolane	Demonstrated to have in vivo activity at siRNA doses as low as 0.01 mg/kg in rodents and 0.1 mg/kg in nonhuman primates.	[99]
L319	di((*Z*)-non-2-en-1-yl)-9-((4-(dimethylamino)butanoyl)oxy)heptadecanedioate)-9-((4-(dimethylamino)butanoyl)oxy)heptadecanedioate	Biodegradable lipid displaying rapid elimination from plasma and tissues, substantially improved tolerability in preclinical studies.	[100]
C12-200	-	Over 95% silencing at a dose of 0.03 mg/kg in non-human primates and 0.01 mg/kg in mice.	[101]
cKK-E12	-	Over 95% silencing at a dose of 0.3 mg/kg in nonhuman primates. Toxicity studies showed that cKK-E12 was well tolerated in rats at a dose of 1 mg/kg.	[102]

### 4.2. Polymeric-Based Nanocarriers

Polymers and their derivatives are an attractive alternative for the development of drug delivery systems. In this regard, polysomes for NA delivery can be formed with an ample variety of materials, including NPs, nano-micelles, dendrimers, hydrogels, and nanoemulsions (Figure 3B) [90]. Some of these developments have even reached clinical stages, and for instance, the natural polymer cyclodextrin enabled the first siRNAs delivery application at the clinical level [11]. Cyclodextrin is a natural macrocyclic oligosaccharide with internal hydrophobic and external hydrophilic structures, which may interact with NAs externally and improve their stability. Nonetheless, a major challenge for a full translation into the clinic is its relatively high toxicity [103,104,105]. Besides cyclodextrin, other natural polymers, including chitosan [106,107], hyaluronic acid [108], dextran [109], and gelatin [110], have been widely studied as promising candidates for NA delivery systems.

Table 2 shows some synthetic polymers employed in NA delivery. PEI (polyethyleneimine) has been demonstrated to be an ideal cationic delivery carrier for NAs [90,111]. Interestingly, block copolymers of PEG-PEI instead of PEI alone have been considered excellent alternatives to decreasing toxicity and improving its performance, mainly due to the ability of PEG to prevent opsonization and avoid specific interaction with blood cells [112]. However, it is important to remark that this property is a function of PEG density and particle size in nanocarriers, as discussed above and demonstrated by Walkey et al. [57] and Han et al. [113]. One of the most important pH-sensitive cationic polymers, pDMAEMA, is widely used for DNA, siRNA, mRNA, and miRNA delivery with acceptable cytotoxicity and combined transfection efficiency [90,114,115,116,117]. Surprisingly, besides serving as delivery vehicles, some polymers also exhibit therapeutic properties [90]. For example, a polymer derivative of the drug metformin, with anti-cancer and anti-diabetic effects, was found to have the ability to deliver siRNA for RNAi therapy without losing its anti-cancer properties [118].

On the one hand, the anti-cancer property of metformin is mainly attributed to the activation of AMP-activated protein kinase (AMPK) [119,120] and inhibition of the mammalian target of rapamycin (mTOR) [121,122]. On the other hand, this polymer’s ability to deliver siRNA may be due to the presence of guanidine groups in its structure. The guanidine has been found to pass through the non-polar membrane of a cell and even across tissue barriers by possibly forming a bidentate hydrogen bond with anionic cell surface phosphate, carboxylates, and/or sulfates on the cell surface [123,124]. As another example, a near-infrared absorbing, dendronized, and semiconducting polymer delivered DNA efficiently and controlled gene expression spatiotemporally together with a heat-inducible promoter [125].

**Table 2 pharmaceutics-13-00428-t002:** Some synthetic polymers frequently used for nucleic acids (NA) delivery.

Abbreviation	Chemical Name	Findings/Relevant Properties	Reference
PEI	Polyethylenimine	The most widely used. It is the organic macromolecule with the highest cationic-charge-density potential.	[90,111]
pDMAEMA	Poly(2-dimethylamino)ethyl methacrylate	Extensively studied and widely used for the delivery of DNA, siRNA, mRNA and miRNA. It had tertiary amines in its structure.	[126]
hDD90-118	-	An hyperbranched poly(beta amino ester) capable of save and effective delivering of mRNA to lung epithelium.	[127]
N5	-	An assembly of poly A binding proteins and cationic polypeptides for enhanced mRNA delivery.	[128]
PAA8k-(2-3-2)	-	A poly(acrylic acid) scaffold grafted with oligoalkylamines promoting enhanced mRNA delivery.	[129]

Other types of polymersomes, such as dendrimers, and polymeric micelles, have proven to be valuable nanocarriers. Dendrimers are highly branched functional polymer-based nanocarriers that comprise an inner core, an amidoamine backbone, and multiple terminal amine groups. Due to this conformation, dendrimers feature plenty of compartment space for loading NAs [90,130,131]. Notably, the coupling of the G0-C14 dendrimer with PEG-PLA polymer leads to cationic amphiphilic dendrimers that have exhibited synergistic anti-cancer effects through the encapsulation of both chemotherapeutic drugs in their hydrophobic interlayer and NAs in their hydrophilic cavity [90,132]. Polymeric micelles, which have received considerable attention in polymer chemistry, are self-assembled from synthetic block copolymers or graft copolymers with an inner hydrophobic core and an outer hydrophilic shell [100]. NAs are more favorably incorporated into micelles’ inner core formed from positively charged polymers through ionic interactions. This enables the incorporation of NAs, such as CRISPR-Cas9 and siRNAs, into polymeric micelles with high stability [133,134]. Davis et al. [135] developed a cyclodextrin-containing polymer to conjugate camptothecin (CPT) with near-neutral zeta potential that properly self-assembles into nanoparticles of about 30 nm diameter. The nanoparticles enter the tumor cells and slowly release the CPT, causing them to disassemble into individual polymer chains that are sufficiently small to be cleared renally. Additionally, the nanoparticles showed long circulation half-lives in animals and humans and targeted localization in tumors. These encouraging results suggest that polymeric micelles can be promising nanocarriers for in vivo and in vitro NA delivery.

Various strategies have been proposed to facilitate endosomal escape through different pathways. For example, cationic polymers (such as PEI) and others with pendant amine groups exhibit a buffering capacity that enables escape from endosomal entrapment through the proton-sponge mechanism and the osmotic lysis effect [96]. By this pathway, protons are pumped by an ATPase into the endosome during its acidification to reach the desired pH to start their maturation. However, amine groups’ presence may produce a buffer effect and sequester the incoming protons because pKa values of such groups are in the range of endolysosomal pH values [136]. Consequently, they can maintain a constant pH, altering the Nernst equilibrium potential. This results in an influx of chloride counterions and water molecules to restore such an equilibrium, thereby producing a pressure increase, which eventually disrupts the endosomal membrane [137] (Figure 4). Another endosomal escape mechanism that pH-responsive polymeric NPs may follow is through particle swelling [138]. As the pH lowers during the endo/lysosome’s maturation process, polymeric NPs tend to swell within the endosomal vesicle. Finally, the endosome’s lysis is reached by either the exerted mechanical strain during swelling or the proton-sponge effect [96,138] (Figure 4).

### 4.3. Inorganic Nanomaterials

Inorganic NPs have become an attractive therapeutic NA and drug delivery approach mainly because they feature several advantages: precise size control, tunable surface properties, and high drug loading efficiency (Figure 3C). [8,90]. Some of the different nanomaterials used for this purpose are presented and discussed below.

#### 4.3.1. Mesoporous Silica

The mesoporous materials are attractive for controlled loading and cargo release due to well-defined pores at the nanoscale, opening opportunities for drug delivery applications. This family of materials contains a plethora of compounds such as alumina, silica, titania, and zirconia. However, mesoporous silica nanomaterials (MSN) have received particular interest in the context of drug delivery [8,139,140,141] mainly due to their excellent physical and chemical stability, high loading capacity, and opportunities for controlled drug release through modifications in surface area, and pore size and form.

The potential use of MSN for the delivery of DNA and siRNA has attracted much attention due to the ease of binding through pre-adsorbed cationic polymers (e.g., PEI) and facilitated endosomal escape by the presence of titrating charged groups capable of membrane destabilization (Figure 4) [73]. However, despite these advantages, PEI-modified mesoporous has proven to induce adverse toxicity effects and lower the amount of siRNA delivered. To overcome this issue, some researchers have attempted to incorporate the polynucleotides into the pores of MSN’s as opposed to their external surface. For example, Kim et al. [142] prepared ultra-large pores by treating MSN with a swelling agent at elevated temperature, which induced a ten-fold expansion in pore size, resulting in both higher DNA loading and transfection efficiency in HeLa cells compared with small pores. In line with this, Li et al. [143] incorporated siRNA into MSN under dehydration conditions. They reported efficient siRNA protection, low cytotoxicity, and cancer cell internalization and subsequent release. This approach allowed a knockdown of both the enhanced green fluorescent protein (EGFP) gene and the B-cell lymphoma 2 (Bcl-2) gene. Similarly, Na et al. [144] demonstrated efficient knockdown of vascular endothelial growth factor (VEGF) gene in vitro and in vivo for MSN with ultra-large pores and negligible cytotoxicity in HeLa cells. This delivery system showed higher gene silencing efficiency than commercially available Lipofectamine 2000, a gold standard for gene transfection.

Some studies have attempted to deliver drugs and siRNA simultaneously. For example, Meng et al. [145] used PEI-coated mesoporous materials to incorporate the chemotherapeutic drug doxorubicin into the porous interior, while the siRNA targeting the P-glycoprotein mRNA in cancer cells was bound to the external PEI-coated surface. As a result, doxorubicin and siRNA’s dual delivery significantly improved doxorubicin’s anticancer effect by silencing the P-glycoprotein expression in a drug-resistant cancer cell line (KB-V1 cells).

MSN has been proposed to design delivery systems for peptide and protein drugs due to favorable properties regarding stabilization and possibilities for release triggering and extended release. For instance, Izquierdo-Barba et al. [146] incorporated the antimicrobial peptide LL-37 and the low molecular weight antimicrobial chlorhexidine into monolithic mesoporous silica. The release of both molecules was slow and controlled by incorporating SH groups in the pore walls. Mesoporous silica-containing either LL-37 or chlorhexidine showed pronounced bactericidal activity against both *Staphylococcus aureus* and *Escherichia coli*. Also, the material containing LL-37 exhibited very low cytotoxicity. MSN-containing LL-37 showed potential for controlling implant-related infections, e.g., for multi-resistant pathogens or for situations where access to the infection site of systemically administered antibiotics is restricted due to collagen capsule formation or other factors.

Additionally, Wang et al. [147] demonstrated that MSN is well-suited for developing protein-based vaccines and concluded that immunological responses vary with particle size and pore characteristics. In this regard, the silica pores proved to successfully entrap and steadily release the model protein antigen bovine serum albumin (BSA). Moreover, oral immunization with the MSN/BSA formulation produced stimulated humoral and mucosal (IgA) responses compared to BSA’s parenteral administration emulsified in Freund’s complete adjuvant. Their findings suggest that the large, honeycombed pores of MSN induced stronger IgG and IgA titers, contributing to developing effective vaccines providing a robust systemic immune response against infectious diseases.

Loading specific functional biomolecules (including peptides and proteins) may enable opportunities for site-localized MSN targeting. Li et al. [148] developed PEI-coated magnetic MSN functionalized with the fusogenic peptide KALA for enhanced green fluorescent protein (EGFP) and vascular endothelial growth factor (VEGF) knockdown. Such a system showed low toxicity and suitable protection of siRNA against degradation. The peptide facilitated the internalization into cells, endosomal escape, and siRNA release into the cytoplasm, leading to an efficient knockdown in tumor cells. In vivo, the intratumoral injection considerably inhibited tumor growth. The conjugation of membrane penetrating peptides, particularly TAT, has also been attempted to reach delivery to the nucleus rather than the cytosol. As Pan et al. [149] reported, TAT-conjugated MSN with a diameter of 50 nm or less effectively targeted the nucleus and showed significant anticancer activity.

#### 4.3.2. Hydroxyapatite and Other Calcium Phosphates

Hydroxyapatite has been highlighted as a drug delivery system mainly due to its excellent biocompatibility. Additionally, it is readily biodegradable, with calcium and phosphate ions as low-toxic degradation products. Moreover, its favorable interaction with bone-forming cells has paved the way for developing many bone regeneration applications [8]. Notably, hydroxyapatite and other types of bone cement have been implemented to deliver growth factors for bone regeneration because they prevent significant conformational changes, chain scission, or protein aggregation. As a result, bioactivity is mostly preserved, as recently reported by several studies describing successful regeneration [8,150].

Also, hydroxyapatite has been investigated as a delivery system for DNA and siRNA. However, it has to be mentioned that aggregation and/or accumulation of hydroxyapatite nanoparticles restrains delivery because DNA precipitation may occur, which amplifies immunogenic response [151]. In this regard, DNA co-precipitation with calcium phosphate is challenging and frequently turns out in bulk calcium phosphate precipitation. Some strategies, such as adjusting the Ca/P ratio, mixing, and selecting the type of calcium phosphate phase, have been proposed to reduce inefficiency and variability [8]. To address the precipitation-induced variability, Sokolova et al. [152] suggested using multi-shell nanoparticles where a calcium phosphate core was coated with DNA. This is followed by another calcium phosphate coating and a second DNA deposition. Results indicate improved transfection efficiency compared to nanoparticles with a single DNA layer.

Chen et al. [153] developed a hydroxylapatite nanorods synthesis that incorporated the stabilizing block copolymer PLGA–mPEG, followed by DNA post-adsorption. They obtained uniform nanorods, 100 nm in length and 25 nm in diameter. In the presence of Ca^2+^, these nanorods promoted a high DNA loading capacity. Comparably, Wang et al. [154] investigated positively charged chitosan-coated hydroxyapatite nanoparticles with lengths of 60–70 nm, enabling efficient DNA loading and showed, under low cytotoxicity, transgene expression in HeLa and NIH3T3 cells comparable to that obtained with the conventional transfection agent lipofectamine. Similarly, Wu et al. [155] researched an in vivo approach based on PEI-modified hydroxyapatite nanoparticles and a recombinant plasmid for enhanced green fluorescent protein (EGFP) expression. Upon administration into the round window membranes of chinchillas, high inner ear transfection efficiencies were demonstrated by the abundant EGFP fluorescence of dark cells in both sides of the crista and around the utricle’s macula.

Yazaki et al. [156] co-immobilized DNA and antibodies within a hydroxyapatite matrix and assessed the obtained materials’ cell-specific gene transfer. They observed the transfection of CD49f-positive CHO-K1 cells in the presence of an anti-CD49f antibody. For an anti-N-cadherin antibody, a corresponding higher transfection was observed for N-cadherin-positive P19CL6 cells but not for N-cadherin-negative UV♀2 cells or P19CL6 cells pre-blocked with anti-N-cadherin.

#### 4.3.3. Layered Double Hydroxides

Layered double hydroxides (LDHs) represent a family of layered materials consisting of positively charged layers with charge balancing anions between them. LDHs are interesting for pH-sensitive drug delivery because of their ability to be dissociated under acidic environments and their significant anionic exchange capacity [8]. Moreover, LDHs may potentially serve as delivery agents for DNA and siRNA. Through ion-exchange of interlayer anions, substantial amounts of polynucleotides can bind to the basal planes with little resistance to nucleotide incorporation from their confining 3D structure [8].

LDHs can be obtained as nanoparticles by adjusting the synthesis conditions. When in contact with mammalian cells, they tend to attach to their membrane and eventually undergo receptor-mediated endocytosis. This process is facilitated by the presence of positive charges on the outer surface of these nanomaterials’ structures. LDH nanoparticles dissolve during endosome acidification, buffering the endosomal pH and facilitating escape into the cytoplasm (via membrane destabilization caused by translocation as depicted in Figure 4) to reach superior transfection efficiency in HEK 293T cells, as evidenced by increased GFP expression as a function of DNA loading [157,158]. Quantitatively, however, delivery efficiency with LDH nanoparticles was low (7–15% of that of the commercial agent FuGENE^®^), which might be attributed to LDH aggregation caused by the long-chain plasmid DNA.

Furthermore, as Wong et al. [159] reported, LDH mediated siRNA delivery efficiencies into cortical neurons and NIH 3T3 cells, varying widely (6–80% and 2–11%, respectively). Such variability was attributed, at least partly, to the ample distribution in the degree of oligonucleotide intercalation. This limited intercalation causes double-stranded nucleic acids to adsorb primarily on the external surface of LDH, where it is prone to enzymatic degradation. To address this issue, recent reports have proposed to prepare smaller LDHs (ca. 45 nm) such that more effective intercalation into these LDH nanoparticles for both dsDNA and siRNA is achievable [160]. This is feasible since the exposed specific surface area increases proportionally with a reduction in the particle size. Then, smaller LDH nanoparticles are expected to have enhanced surface adsorption of the anionic dsDNA/siRNA, facilitating, therefore, their intercalation. The vehicles with dsDNA/siRNA were transfected into HEK 293T cells and demonstrated efficient silencing.

Nonetheless, larger LDH vehicles have also demonstrated efficient transfection and gene silencing. For example, Li et al. [161] developed a delivery system for DNA vaccination that consisted of an LDH/DNA complex with an average diameter of 80–120 nm, which showed a high GFP transfection efficiency in vivo. Furthermore, intradermal delivery of pcDNA3-OVA/LDH in C57BL/6 mice caused an antibody response significantly higher than that of naked DNA. They also proved enhanced immune priming and protection from tumor challenge in B16-OVA melanoma model tumors.

#### 4.3.4. Lanthanide Upconversion Particles

Lanthanide-doped upconversion nanoparticles (UCNPs) follow a photochemical internalization to achieve endosomal rupture and subsequently delivery of carried biomolecules as schematized in Figure 5. UCNPs are typically composed of trivalent lanthanide dopant ions embedded in a host lattice, which is expected to match dopant ions to yield low phonon vibration energies and excellent chemical stability [8]. Fluorides and oxides have been the most commonly used host lattices for Ln-doped UCNPs. Particularly, fluoride-based (e.g., NaYF4) UCNPs have been identified as an efficient alternative due to their low phonon vibration energy [8]. The luminescence efficiency of Ln-doped UCNPs has been improved by incorporating two types of dopant ions. One of them emitting visible light (activator) while the other acting as an energy donor (sensitizer) (Figure 5). Yb^3+^ is usually selected as a sensitizer due to its high absorption coefficient and upconversion efficiency, while Er^3+^, Tm^3+^, and Ho^3+^ are generally the choice for activators [8]. The emerging interest in UCNPs arises from better tissue penetration by near-infrared (NIR) compared with visible light excitation, thereby enabling deep tissue applications. Moreover, UCNPs show excellent stability against photobleaching and photochemical degradation, making them attractive for a range of biomedical applications, including cancer therapy, bio-labeling, fluorescence, magnetic resonance imaging, and drug delivery [162,163].

UCNPs have also been applied as siRNA nanocarriers. For example, Jiang et al. [164] prepared silica-coated NaYF4 nanoparticles co-doped with Yb/Er and conjugated them with folic acid and anti-Her2 antibody for cell targeting. Such nanoparticles exhibited efficient intracellular uptake, resulting in the silencing of luciferase expression. Other research conducted by Yang et al. [165] reported silica-coated UCNPs with cationic linkers as carriers for siRNA delivery. After cellular uptake, these UCNPs emitted UV light-induced by NIR irradiation, followed by siRNA photorelease. Released siRNA showed to conserve its biological activity as corroborated by successful gene silencing tests.

Furthermore, Guo et al. [166] proposed silica-coated and amino silane-modified UCNPs as low cytotoxicity vehicles for the foot-and-mouth disease virus (FMDV) DNA vaccine, reporting a transfection efficiency similar to lipofectamine. Immune responses observed after intramuscular administration included T-lymphocyte proliferation and neutralizing antibodies (anti-FMDV specific antibodies). After challenging experiments, all vaccinated guinea pigs were demonstrated to be fully protected from FMVD.

#### 4.3.5. Gold Nanoparticles

Besides their use in nanomedicine diagnostics and biosensors, gold nanoparticles (AuNPs) provide exciting possibilities for drug delivery. Mainly due to these nanoparticles’ small size, large loads of biomacromolecular drugs may be readily adsorbed at their surface [8]. Some alternatives for the release of these drugs include simple desorption induced by a change in pH or ionic strength, light exposure for drugs covalently bound to the nanoparticles through photolabile linkers, or thiol reduction linkages used for drug chemisorption [8]. AuNPs have received massive attention as potential carriers for DNA and siRNA, as both may be either physisorbed to the gold nanoparticles or bound covalently through thiol bonds. Both approaches usually incorporate some passivating surface modification to colloidally stabilize them and/or promote endosomal escape (Figure 4) [8,167]. Early studies employed AuNPs coated with SH–PEG5000–PAMA7500 (PAMA: poly(dimethyl aminoethyl methacrylate)) and SH–siRNA, reaching a loading of 45 siRNA molecules per AuNP and 65% knockdown of luciferase expression in HuH-7 cells [168]. Following a similar approach, a recent report attempted to coat AuNPs with a combination of SH–siRNA and a shorter PEG (SH–PEG400), which led to efficient luciferase knockdown in HeLa cells [169].

Another approach attempted to initially coat the AuNP surface, e.g., with SH–PEG–NH2, and then attach a siRNA molecule using a disulfide crosslinker conjugated to the terminal amine as disulfide bonds are easily cleavable in reductive environments such as that of the cytoplasm. Akinc et al. [170] applied this strategy to obtain a DNA loading of 40 duplexes per particle, which were subsequently coated with different poly(β-amino esters) (PBAEs) for enhanced cellular uptake and endosomal escape (Figure 4). Such modification proved to reduce luciferase expression in HeLa cells considerably. Instead of this plan of action, siRNA may be simply physisorbed on AuNPs, considering that most AuNPs in aqueous solutions are prepared via citrate reduction, which renders a superficial negative charge. Thus, surface modification by cationic polymers or lipids could be adopted for promoting siRNA adsorption on AuNPs via layer-by-layer (LbL) deposition [8]. The application of LbL approaches via PEI as positively charged polymer has shown pronounced differences compared to particles terminated with siRNA regarding cell distribution and gene silencing, thereby pointing to the importance of endosomal escape [8,171].

Although PEI has been employed in considerably higher concentrations to adapt AuNPs as DNA carriers [172], its toxicity has given rise to the search for alternative cationic polymers with reduced toxicity and additional functional features. Guo et al. [173] synthesized a system with both PEI and cis-aconitic anhydride-functionalized poly(allylamine) (PAH-Cit) for siRNA complexation. Under acidic conditions (such as those found in endosomes), PAH-Cit undergoes charge reversal, resulting in a complicated disassembly and siRNA release. Through an LbL system, they reported 80% silencing of lamin A/C protein expression, which was about four times higher than that observed for complexes formed with a non-charge reversing polymer. In parallel, confocal microscopy imaging confirmed increased endosomal escape for the complexes containing PAH-Cit.

Similarly, Han et al. [174] proposed chitosan to reduce and stabilize AuNPs, achieving positively charged AuNP-CS core particles, in which PAH-Cit/PEI and siRNA were sequentially deposited by electrostatic interaction. Cytotoxicity of the resulting particles against HeLa and MCF-7R cells was found to be insignificant, along with efficient protection of siRNA against enzymatic degradation. In vitro release was triggered under acidic conditions because of the charge-reversal of PAH-Cit. In vivo, pH-dependent siRNA release facilitated the endosomal escape. In drug-resistant MCF-7 cells, specific gene silencing of the drug exporter P-gp was observed upon doxorubicin’s uptake.

LbL modification of AuNPs has also been utilized for target-specific intracellular delivery of siRNA. Lee et al. [175] reported a system composed of cysteamine (CM)-modified AuNPs, PEI, and hyaluronic acid (HA). In this case, the HA was incorporated into the system to avoid nonspecific binding to blood serum components, which, in turn, reduces the reticuloendothelial system (RES) uptake. This results in increased bioavailability and lower dose-limiting side-effects. Additionally, hyaluronic acid facilitates the targeting of the CD44 hyaluronan receptor. Apart from reduced cytotoxicity, such a system led to 70% gene silencing in the presence of serum. Furthermore, target-specific intracellular delivery of AuCM/siRNA/PEI/HA to B16F1 cells through HA receptors was corroborated through a competitive binding assay with free HA. Lu et al. [176] targeting of siRNA-containing AuNPs has also been demonstrated for a vehicle consisting of gold nanoshells (AuNSs) coated with SH siRNA, and TA–PEG–F. Thioctic acid (TA) and folic acid (F) provided attachment to the gold surface and improved targeting capabilities, respectively. The targeting of HeLa cells has also been demonstrated for nanoparticles functionalized with folate molecules.

Broader insight to address targeted silencing by AuNP-carried siRNA has been reported. As an example, Conde et al. [177] used a multifunctional approach, where several biochemical moieties, including cell-penetrating and cell adhesion peptides, were selected to improve cellular recognition and siRNA uptake. Another divergent pathway to achieve endosomal escape of siRNA-containing gold nanoparticles has been proposed by Braun et al. [178], who developed AuNSs susceptible to local heating when irradiated with NIR light, whose mechanism acts similar to that schematized in Figure 5. These nanoshells were PEG-modified, followed by further conjugation with the cell-penetrating TAT peptide to promote cell uptake. A reduction of 80% in GFP expression was observed in C166 mouse endothelial cells compared to control cells treated with non-targeting complexes and in the absence of NIR irradiation. Notably, in the absence of irradiation or low irradiation power, siRNA/AuNP complexes show no silencing, which demonstrated the importance of light-mediated activation of AuNPs for their cellular uptake and distribution. In parallel, confocal microscopy images indicated de-complexation for a lower power of irradiation, but the endosomal escape of siRNA only at high-intensity irradiation. Analogously, Huschka et al. [179] attached poly-L-lysine to AuNS, followed by electrostatic binding of either DNA or siRNA, which was eventually released upon laser irradiation at 800 nm (near the nanoshell’s resonance wavelength), resulting in ~50% downregulation of GFP. By changing the shape of gold nanoparticles to gold nanorods, NIR optical responses are significantly improved. Oyelere et al. [180] demonstrated this by conjugated gold nanorods with the SV40 virus NLS peptide for nuclear targeting.

Although most reported studies are focused on delivering siRNA and DNA by AuNPs, some works have attempted to deliver proteins and peptides with them. Lee et al. [181] proposed using AuNPs conjugated with thiol-modified hyaluronic acid as a vehicle for interferon R (IFNR), providing an alternative to INFR–PEG conjugates for the treatment of hepatitis C. The AuNP/HA/IFNR complex showed similar biological activity to the PEG–IFNR conjugate and a considerably enhanced serum stability, which lasted even seven days after injection in murine liver tissue. Additionally, high stability was confirmed by the absence of either non-conjugated IFNR or the PEG-conjugate. Moreover, Verma et al. [182] reported a cationic ammonium-functionalized AuNP/anionic β-galactosidase complex that retained protein activity after triggered release by glutathione reduction. Similarly, Liu et al. [183] found conserved insulin bioactivity after interaction with AuNPs [68], while Paciotti et al. [184] confirmed promising results for AuNP conjugated with thiolated PEG tethering the tumor necrosis factor.

#### 4.3.6. Magnetic Nanoparticles

Magnetic nanoparticles are referred to as iron oxide nanoparticles (IONPs), are magnetic Fe_3_O_4_ or Fe_2_O_3_ nanocrystals that may interact with external magnetic fields, enabling several applications in nanomedicine, e.g., contrast agents in magnetic resonance image (MRI), magnetic hyperthermia therapies, or magnetically triggerable drug delivery systems [8,185]. IONPs smaller than 20 nm may be magnetized in the presence of an external magnetic field but lose such properties when the magnetic field is removed. This attribute is called superparamagnetism and is lost for nanoparticles, with a size of over 30 nm [8]. Another strategy to maintain superparamagnetism and simultaneously reach a more robust magnetic response consists of assembling superparamagnetic INOPs (SPIONs) into a large matrix-forming material. For example, magnetic nanocrystals can be assembled aided by polymeric cross-linkers such as poly(acrylic acid) or PEI. This approach allows obtaining particle clusters with a size over 100 nm that exhibit even higher magnetic moments than 8 nm-sized nanocrystals [185]. For biomedical applications, IONPs/SPIONs must be surface modified to improve their colloidal stability and diminish the biological system’s adverse effects. Some surface modification strategies include chemical functionalization (e.g., silanization), coordination binding (e.g., catechol binding), and conjugation of hydrophilic polymers, such as PEG, PEI, and poly(vinyl alcohol) (PVA), or natural polysaccharides such as dextran, heparin, and chitosan [8].

For example, Al-Deen et al. [186] proposed a potential malaria DNA vaccine system based on PEI and hyaluronic acid-modified SPIONs. In many other polyelectrolyte systems, particle size, charge, stability, and DNA binding/release were influenced by the mixing order. The stabilization of DNA from enzymatic degradation was also achieved by controlling these factors. Moreover, Shah et al. [187] developed a system consisting of magnetic nanoparticles modified by oleic acid and coated with thermoresponsive poly(N-isopropyl acrylamide-co-acrylamide). The resulting composite particles displayed a magnetic core of about 18 nm and a shell thickness of about 13 nm.

Interestingly, the polymeric shell is extended and hydrated under temperatures below its lower critical solution temperature (LCST; 39 °C), while a substantial de-swelling of the polymeric shell occurs at the LCST, triggering a notable “squashing release” of the incorporated drug (doxorubicin). Heating is produced when these nanoparticles are exposed to an oscillating magnetic field, which favors a localized and magnetically triggered drug release. Despite the functional benefits obtained with polymeric surface coatings, some physical properties may be negatively affected, such as the size and/or nanoparticles’ morphology. As shown in Cormode et al. study [188], issues regarding efficient nanoparticle targeting to hepatocytes can be addressed with the aid of different polymer surface modifications. In their work, IONPs were firstly encapsulated with oleic acid and secondly by an amino-substituted polymer. Next, these nanoparticles were coated with PEG to reduce opsonization and DNA binding efficiency was monitored for an increasing fraction of PEG. Nanoparticles containing less than 5% PEG bound DNA efficiently, while this was either suppressed or prevented for particles with 10% and 25% PEG. Thus, although PEG coating promoted a decrease in opsonization, the transfection efficiency was reduced. However, promising functional performance may be obtained from balanced PEG-containing SPIONs, as reported by Kenny et al. [189]. They used PEG-modified SPIONs for siRNA delivery to reduce opsonization and RES uptake. This resulted in increased bloodstream circulation and, ultimately, the preferential uptake by cancer tissue through the enhanced permeability and retention (EPR) effect. Magnetic resonance imaging (MRI) analyses demonstrated that these nanoparticles accumulated in xenograft tumors after intravenous injection. Simultaneously, the combination of MRI and fluorescence microscopy corroborated tumor co-localization of siRNA and nanoparticles. Consequently, the delivery of the anti-cancer siRNA by these carriers resulted in a significant tumor growth reduction.

PEI had been prevalently used with magnetic nanoparticles to deliver a malaria DNA vaccine [190]. By taking advantage of the magnetic responsiveness of SPIONs, DNA-containing nanoparticles may be pulled to and through cell membranes by applying external magnetic fields or magnetic field gradients, which led to an increase in transfection efficiency. Similarly, Zhang et al. [191] developed a system consisting of yolk-shell nanocapsules (NCs), core-shell nanostructures with a movable magnetic core, surrounded by interstitial hollow space, and a SiO_2_ shell. The system was finally coated with PEI for the magnetically assisted delivery of b-actin siRNA into HeLa cells. Their results demonstrated a significant gene silencing without major cytotoxic repercussions. Comparably, Liu et al. [192] developed dendrimer-modified IONPs for DNA delivery. Such IONPs were mixed with plasmid DNA at a ratio resulting in a negative charge, followed by PEI modification. Unlike nanoparticle-free PEI/DNA complexes, the DNA/SPION systems showed improved transfection efficiency for HeLa, COS-7, and 293T cells in the presence of an external magnetic field. These remarkable results could be attributed to the enhanced cellular uptake promoted by the magnetically assisted direction of the functionalized nanoparticles to cell surfaces.

SPIONs, when combined with external magnetic fields, may contribute to developing intracellular protein- and peptide-delivery systems. For example, Veiseh et al. [193] explored the peptide chlorotoxin delivery, finding that SPION-bound chlorotoxin showed increased uptake in glioma cells compared to free chlorotoxin. Moreover, Chertok et al. [194] developed a SPION-mediated protein delivery system inserted in brain tumors, using heparin-coated iron oxide nanoparticles modified with PEI before binding the model protein β-galactosidase. In a 9L rat intracerebral glioma model, following carotid artery injection, such protein nanocarrier was proven to accumulate at tumor sites in the brain, which was further exacerbated in the presence of an external magnetic field.

#### 4.3.7. Carbon-Based Nanostructures

Carbon nanotubes, graphene-derived materials, and quantum dots are part of a family of carbon-based nanostructures that, besides their nanomedicine application as biosensors, have been reported as promising alternatives for delivery systems [8].

Carbon nanotubes (CNTs) themselves are colloidally unstable in aqueous solution, flocculating to thick and inhomogeneous bundles, resulting in considerably toxic repercussions, such as acute pulmonary toxicity induction of inflammatory reactions and granulomas [8,195,196]. However, the functionalization of CNTs significantly contributes to reducing inflammation after subcutaneous administration and a good tolerance following *i.v.* and *i.p.* injection and oral administration. Surface modification of CNTs is, therefore, of paramount importance for their biomedical application.

Despite the low extent of scientific clarity about the clearance mechanism of CNTs, several reports have indicated that, after *i.v.* administration, functionalized CNTs accumulate in the reticuloendothelial system (RES). As for other colloidal drug carriers, functionalized CNTs are usually internalized via endocytic pathways. However, the type of mechanism appears to depend on the CNT length. Kang et al. [197] revealed that 100–200 nm single-wall nanotubes (SWNTs) are taken up through clathrin-coated pits, whereas both clathrin-coated vesicles and the caveolae pathway are the preferred mechanisms for shorter SWNTs (50–100 nm).

Some early reports indicated that amine-functionalized CNTs enhanced binding DNA yet led to low transfection efficiency [198]. To facilitate further DNA binding, cell uptake, and endosomal release, PEI has been used to modify the CNT surface due to its protein-sponge properties (Figure 4) [199]. Addressing nucleotide-binding and condensation, Liu et al. [200] developed a system in which a chitosan derivative containing β-cyclodextrin and pyrene was used to functionalize CNTs. This approach led to improved DNA binding and condensation due to the cooperation between the cationic charges of chitosan and the aromatic nature of the pyrene groups, as evidenced by atomic force microscopy and dynamic light scattering.

Endosomal escape and intact release to the cytosol are critical requirements for DNA and siRNA delivery. To address these issues, Kam et al. [201] prepared a short SWNT surface modified by poly(ethylene glycol)-modified phospholipids containing terminal amine or maleimide groups. Thiol-modified DNA or siRNA were then linked through cleavable disulfide bonds to enable enzyme-mediated release from endosomal and lysosomal compartments (Figure 4). Due to the remarkable siRNA binding efficiency and the presence of cleavable disulfide links, the siRNA cargo achieved high endosomal escape and, consequently, gene silencing efficiencies on human T cells that exceeded those of several benchmark transfection agents. Furthermore, shortened CNTs functionalized with lipids and natural amino acid-based dendrimers were tested in vivo by McCarroll et al. [202]. These modified CNTs allowed a systemic delivery of siRNA, resulting in about 50% silencing of apolipoprotein B (ApoB) in mice liver. This translated into plasma level reduction of ApoB above 60%, consequently alleviating plasma cholesterol, with no observable adverse effects.

Carbon nanotubes have also been used for protein and peptide delivery, providing benefits such as improved stability related to proteolytic degradation [8,195]. For example, Villa et al. [203] modified CNTs by chemical conjugation of peptide antigens at high density through bis-aryl hydrazine to enhance immune responses for weakly immunogenic peptides. These nanotubes showed to be internalized by macrophages and dendritic cells. Notably, BALB/c mice immunized with peptide-modified CNTs exhibited specific IgG responses against the peptide, while the peptide alone showed none. Moreover, they observed negligible toxicity in vitro and no adverse antibody responses in vivo.

Graphene behaves similarly to carbon nanotubes. Unmodified graphene nanosheets may produce undesirable biological responses [204]. In contrast, graphene oxide (GO) is usually employed as a drug carrier, frequently after additional surface modification, e.g., by π–π-stacking, chemical conjugation, or physical adsorption [8,205]. Recent research developments have proposed the use of polymer-modified GO for gene transfection. For this purpose, Feng et al. [206] modified GO with PEI of two molecular weights (1.2 and 10 kDa). The obtained conjugates showed good colloidal stability at physiological conditions and attenuated toxicity. Furthermore, both systems were subsequently loaded with plasmid DNA for EGFP in HeLa cells. GO-PEI (1.2k) showed increased EGFP expression, while the PEI without GO resulted in ineffective EGFP transfection. GO’s presence and absence showed no impact on the EGFP transfection efficiency for the higher molecular weight PEI. However, the GO-PEI complex exhibited lower toxicity than PEI-10k.

Several authors have proposed GO-based delivery systems containing targeting groups to enhance the outcome of nucleotide delivery in terms of transfection or gene silencing. For example, Ren et al. [207] developed a GO-PEI system functionalized with the nuclear signal peptide PKKKRKV (PV7) either after forming the GO-PEI/DNA complex or simultaneously with GO-PEI and pDNA. In vitro transfection demonstrated that the simultaneous addition of GO-PEI, DNA, and PV7 led to complexes promoting higher transfection efficiency than either GO-PEI or PEI alone. Also, PV7 was found to enable GO-PEI delivery of DNA into the nucleus. However, this complex showed higher cytotoxicity for 293T and HeLa cells than the GO-PEI system. In a similar strategy for cell-selective uptake in cancer therapy, Yang et al. [208] developed GO-PEI complexes containing folate groups for the targeted delivery of hTERT siRNA, resulting in efficient uptake in HeLa cells, as well as silencing of protein expression.

In addition to DNA and siRNA delivery, GO has been explored to develop protein- and peptide-delivery systems, as demonstrated by Shen et al. [209]. They proposed a system consisting of GO functionalized with amine-terminated hexafunctional PEG, which allowed physical adsorption of proteins. Consequently, the resulting system was proven to protect the adsorbed proteins from enzymatic degradation, thereby enabling effective cytosol protein delivery. Thus, a pronounced loss of cell viability was obtained after delivery of ribonuclease A (RNAse), while cell growth was promoted by delivering protein kinase A (PKA).

Fullerenes are one of the pioneer nanoparticles to be proposed to develop more efficient delivery systems [210]. They, referred to as Buckminster fullerenes or Buckyballs, are among the allotropes of the carbon family of nanomaterials [211]. Besides their specific geometry, size, and surface characteristics, fullerenes possess a structure consisting of sp2 carbons that confer unique chemical and physical properties [212,213]. C60 is the most abundant fullerene in the synthesized composition [214]. Among several advantageous properties of this compound, its dual behavior in producing and downregulating reactive oxygen species (ROS) opens the possibility for an adaptive response according to the application. For example, C60 may produce oxygen species when exposed to visible light, which gives the possibility to use it for photodynamic therapy. Alternatively, it downregulates ROS in other cases, which can be utilized as a neuroprotective agent [215]. However, the mechanism of this action is still unknown and requires further investigation. The use of fullerenes in biological applications is still limited due to their poor water solubility and the need for organic solvents for their synthesis. Several strategies such as the preparation of two-phase colloidal solutions, synthesizing fullerene derivatives, fullerene polymers, encapsulation in special carriers (e.g., cyclodextrins, calixarenes, polyvinylpyrrolidone, micelles, liposomes, etc.), chemical modification [by adding hydrophilic substances such as amino acids, carboxylic acids, polyhydroxy groups (fullerenols) and amphiphilic polymers], among others have been proposed to improve its hydrophilicity and water solubility [213].

Fullerene-based nanovehicles have been used for siRNA delivery. For example, Wang et al. [216] conjugated terminally aminated dextran to C60. Next, the conjugate was further positively charged by covalently linking ethylenediamine to the dextran. The cytotoxicity, cellular uptake, intracellular distribution, and in vitro RNA interference (RNAi) of the siRNA/C60-Dex-NH2 complex was evaluated in the human breast cancer cell line MDA-MB-231. Also, the RNAi efficiencies mediated by C60-Dex-NH2 in vivo were assessed in subcutaneous tumor-bearing mice. This complex showed a specific amphiphilic skeleton forming micelle-like aggregate structures in water, preventing siRNA from being degraded by ROS. Moreover, after exposure to visible light, the complex could trigger controllable ROS generation, which could destroy the lysosome membrane, promote the lysosomal escape, and enhance the gene silencing efficiency of siRNA in vitro (up to 53% in MDA-MB-231 cells) and in vivo (up to 69% in tumor-bearing mice).

#### 4.3.8. Quantum Dots

Quantum dots (QDs) are nanocrystals composed of semiconductor materials, exhibiting attractive photophysical properties, including high quantum yield, resistance to photobleaching, and tunable photoluminescence, which have attracted significant attention as potent tools for biomedical applications [8,217,218]. However, toxic effects shown by several QDs have restricted their potential for biomedical purposes. Thus, as for carbon nanotubes and graphene-derived materials, some surface modifications, particularly surface coatings, have been proposed to attenuate these undesired features [8].

QDs have been widely explored for siRNA delivery [219]. As an example, Yezhelyev et al. [220] developed CdSe QDs coated with tertiary amine-carboxylic acid proton sponge coatings, allowing endosomal escape from acidifying endosomes, which led to a significant improvement in gene silencing for MDA-MB-231 cells, with a simultaneous lowering in toxicity compared to benchmark transfection agents. Another analogous strategy proposed by Qi et al. [221] investigated real-time imaging and delivery of siRNA by amphipol-coated QDs, leading to effective silencing and considerably reduced toxicity in serum-free and complete cell culture media compared to lipofectamine and PEI, which are considered benchmark siRNA carriers. Moreover, CdSe/ZnSe core-shell QDs were functionalized by arginine-modified β-cyclodextrins (β-CDs), improving colloidal stability in cell culture media, as well as reduced cytotoxicity, without decreasing quantum yield significantly. Interestingly, QDs with positive side chains containing amino acids were found to be readily internalized. Binding of siRNA to the β-CD-Arg–QD resulted in good protection from siRNA degradation in calf serum. Additionally, silencing efficiency for the HPV-18 E6 oncogene in HeLa cells was above 80%.

### 4.4. Cell-Penetrating Peptides

Cell-penetrating peptides (CPPs) are attractive nanocarriers for the delivery of nucleic acids and proteins. The first application of CPPs was in the delivery of nucleic acids to cells through electrostatic interactions [222]. In general, the delivery of nucleic acids through this approach offers numerous advantages such as protection of cargoes from degradation, effective internalization into specific target cells, improved release of cargoes intracellularly either at the cytoplasmic level (e.g., antisense nucleotides, RNAi therapies) or the nucleus (e.g., plasmid DNA), high biological activity at low doses, negligible cytotoxicity, and good biosafety for therapeutic studies in vivo [223].

Several CPP-based conjugates have been synthesized and tested for the delivery of siRNA. For example, Kumar et al. [224] developed siRNA delivery nanosystems for the central nervous system based on a small peptide derived from the rabies virus glycoprotein (RVG, a ligand for acetylcholine receptor) modified with polyarginine (Arg9). In vitro studies showed effective gene silencing and protection against the fatal viral encephalitis in a mouse model. Eguchi et al. [225,226] produced a nanovehicle composed of a TAT fusion protein and a double-stranded RNA-binding domain to efficiently deliver epidermal growth factor receptor (EGFR) and AKT serine/threonine kinase 2 (Akt2) siRNAs to intracranial glioblastoma tumors in a mouse model. In parallel, non-covalent approaches allowed developing stable complexes of CPPs to deliver siRNA. Indeed, the first non-covalent approach enabled the production of complexes with the MPG peptide (derived from the hydrophobic fusion peptide of HIV-1 gp41 plus the hydrophilic NLS of SV40 large T antigen) [227]. These complexes facilitated the delivery of siRNAs targeting OCT-4 into mouse blastocytes and subsequently silencing cyclin B1 (a cell cycle regulator) to reduce cell differentiation and proliferation, respectively [228]. Moreover, an amphipathic CPP named *Cady*, containing arginine and tryptophan residues effectively formed stable complexes with siRNA to efficiently achieve gene silencing in both suspension and cell lines such as human osteosarcoma U2OS, THP1 monocytes, human umbilical vein endothelial and mouse 3T3C cells [229].

To significantly improve siRNA delivery systems’ potency, researchers have proposed the stearylation of CPPs. For instance, a stearyl-TP10 analog modified with trifluoromethylquinoline was used to increase endosomal escape and effective siRNA delivery in Jurkat cells and human umbilical vein endothelial cells (HUVEC) [230]. Similarly, the STR-KV peptide (stearyl-HHHKKKVVVVVV) complexed with siRNA targeting the glyceraldehyde-3-phosphate dehydrogenase (GAPDH) exhibited 80–87% gene silencing efficiency and low cytotoxicity [231]. These studies showed that stearylation of CPPs holds a significant promise as a novel alternative to increase the efficiency in siRNA delivery systems.

The use of CPPs also has demonstrated to facilitate the intracellular delivery of diverse proteins and peptides. For example, a system composed of β-galactosidase linked to the TAT peptide exhibited improved blood-brain barrier penetration after intraperitoneal administration [232]. Other advances have shown the effective delivery of anti-apoptotic proteins into cells through their conjugation to CPPs. For instance, Cao et al. [233] obtained protective effects in neurons of a murine middle cerebral artery occlusion model by conjugating the Bcl-xL protein to the TAT CPP. Similarly, a peptide inhibitor of the apoptotic protease-activating factor (Apaf-1) was modified by its conjugation to the CPPs penetratin and Tat. Both CPPs enhanced cellular uptake, but the penetratin conjugate was more effective at inhibiting apoptosis, likely due to the Tat conjugate’s higher cytotoxicity [234]. CPP-mediated delivery of peptides and proteins has mainly been implemented to address cell penetration and targeting to tumors. For example, p53-derived peptides conjugated with the TAT or polyarginine peptides were injected into a peritoneal carcinomatosis mouse model with increased mice survival results [235]. Additionally, a complex composed of a peptide inhibiting casein kinase 2 (P15) activity was injected in mice. to promote enhanced anti-tumor effects [235]. The delivery of proteins and peptides may potentially be improved by their conjugation to CPPs. Consequently, this might provide a green light to developing a more comprehensive variety of nanovehicles concerning the treatment of different types of malignant diseases.

## 5. Enhancing Endosomal Escape of Nanocarriers by Conjugating Cell-Penetrating Peptides

Since genetic material fails to cross the biological membranes unaided, the use of cell-penetrating peptides (CPPs) have been widely implemented for the delivery of nucleic acids [236,237], proteins [238] and drugs [239]. A CPP typically consists of about 30 or fewer amino acids, whose sequence determines their capability to penetrate the cell membrane [240]. The most widely reported membrane penetration mechanisms include direct penetration, endocytosis, and transitory membrane pore formation [241]. The exact mechanism of CPP action depends on the nature of the cargo, cell type, membrane composition, and peptide concentration [241,242]. Cationic CPPs enhance the delivery of therapeutic molecules by interaction with negatively charged domains of the plasma membrane. Some CPPs have different mechanisms to facilitate penetration of cell membranes, including increasing the cell membrane’s fluidity and forming transient lamellipodia to transport cargoes across the membrane [241,243]. The conjugation of CPPs to proteins and genes for delivery applications can help them escape from the reticuloendothelial system while avoiding enzymatic degradation and achieving high nuclear localization [244]. Thus, the delivery of genes and therapeutic agents can be better achieved by combining CPPs with non-viral delivery systems, including nanoparticles, micelles, liposomes, among others.

### 5.1. Nanoparticles

Several studies have developed delivery systems with enhanced endosomal escape by conjugating CPPs on the surface of different types of nanoparticles. The mechanism followed by these systems is shown in Figure 4. For example, Suarez-Arnedo et al. [245] immobilized the antimicrobial CPP Buforin II (BUF2) on magnetite nanoparticles to enhance their cell penetration. BUF2-magnetite nanobioconjugates were capable of bypassing mammalian and bacterial membranes very effectively without promoting significant disruption. Nonetheless, after antimicrobial assays, it was found that BUF2 lost antimicrobial activity, most likely due to blockage of the residues involved in abrogating the replication machinery of *E. coli*. Another similar approach was proposed by López-Barbosa et al. [246], where they conjugated the Outer membrane protein A (OmpA) on magnetite nanoparticles. The OmpA is a member of the outer membrane protein family that possesses cell-penetrating properties. The obtained nanobioconjugates could translocate model lipid bilayers of egg-lecithin liposomes and caused an effective disruption when subjected to magnetic forces. Furthermore, confocal microscopy confirmed the entry of immobilized nanoparticles into THP-1 cells’ cytoplasm, reaching high endosomal and lysosomal escape levels. Taken together, these findings might open novel perspectives toward a new family of gene and drug delivery systems.

Drug resistance is an emerging challenge for the treatment of cancer [247]. To address this issue, the delivery of gene therapies such as siRNA has been considered as a promising strategy to either silence defective genes or down-regulate the proteins causing drug resistance [248]. Therefore, by combining the regulation power of siRNAs with the ability to facilitate cell entry from CPPs, it is possible to develop delivery systems to enhance the penetration and bioavailability of drugs in tumors. For example, the breast tumor cell-penetrating peptide PEGA-pVEC and hyaluronic acid as a targeting media were co-embedded in mesoporous silica nanoparticles to prepare the novel cascaded targeting nanoparticles (HACT NPs). Delivery of siRNA along with a chemotherapeutic drug for breast cancer treatment proceeded from a rattle mesoporous silica system. The nanostructured systems accumulate at tumor vasculature and are captured by PEGA-pVEC mediated endocytosis. The hyaluronic acid’s presence facilitates their targeting and avoids drug leakage until the enzyme hyaluronidase fully degrades the nanoparticles. The siRNA and drug are then released in a controlled manner to silence the gene causing drug resistance and induce the combined therapeutic effect [249]. Other studies proposed using electrostatically stable TAT modified mesoporous nanoparticles for co-delivery of doxorubicin (DOX) and vascular endothelial growth factor (VEGF) siRNA. These nanostructures are composed of an anionic layer consisted of poly (allylamine hydrochloride)-citraconic anhydride (PAHCit), and a cationic outer layer consisted of galactose-modified trimethyl chitosan-cysteine (GTC) to entrap siRNA. This system enabled the release of siRNA intracellularly to elicit silencing of target mRNA and promoted sustained release of doxorubicin at the nucleus. The cytoplasmic glutathione (GSH) caused cleavage of the disulfide bond and the subsequent release of siRNA. This vector efficiently delivered DOX and siRNA and resulted in enhanced cytotoxicity and anti-angiogenesis, critical in anticancer treatments [250].

In other approaches, the condensation of negatively charged plasmid DNA (pDNA) with positively charged TAT peptide served to synthesize delivery nanoplatforms. These vectors were prepared at various (N/P) ratios, representing nitrogen groups from cationic peptides and the phosphate groups from pDNA molecules. Additionally, a cationic surfactant alkyltrimethylammonium bromide (CnTAB) was added to help compact and package the pDNA. Chemotherapeutic agents could be loaded into the pDNA/TAT/surfactant complex and the release behavior of pDNA was controlled by varying the N/P ratio. These nanocomplexes offer a platform for co-delivery of anticancer drugs and pDNA with enhanced cytotoxic effects that combat drug resistance [251]. Similarly, a co-delivery vector was synthesized by conjugating TAT-PEG-PEI-Oleic acid for co-delivery of pDNA and docetaxel. The results showed that the obtained TAT-pDNA-DTX based lipid nanoparticles could work as a promising nanoplatform for the co-delivery of chemotherapeutic drugs and genes with improved transfection and therapeutic efficacy [219].

The application of cell-penetrating peptides have also gained attention for the co-delivery of proteins and drugs to combat the multidrug resistance of several cancer types. Silver nanoparticles were coated with albumin to encapsulate the antihelmintic drug Albendazole whose main action is to inhibit tubulin polymerization. Trichosanthin (TCS), a protein with antitumor activity but lacking specificity and having poor uptake and a short half-life, was bound to these negatively charged nanoparticles. The nanoplatforms were further modified with protamine, a cell-penetrating peptide, to form a stable nano-system. This co-delivery system inhibited tumor cell proliferation by cell cycle arrest and increased apoptosis rate. In addition, an anti-MDR effect was observed in the A549/T lung tumor mouse model [252].

### 5.2. Micelles

CPPs have also been conjugated to polymeric micelles. For example, a polymeric system consisting of poly(ε-caprolactone)-PEG and poly (histidines) PCL-PEG-PHIS was bound to a CPP with the sequence CKRRMKWKK. These micelles were used for co-delivery of VEGF siRNA and paclitaxel. The micelles showed efficient gene transfection and anti-proliferative effects in MCF-7 cell lines. Moreover, in vivo studies in a mouse model demonstrated enhanced apoptosis, reduced VEFG expression, and, therefore, decreased tumor growth [253].

The use of lipid-modified cell-penetrating peptides has been proposed for the treatment of hepatic carcinoma. Micelles formed by self-assembly of lipid-modified CPPs were loaded with the AMPK activator (AMP-activated protein kinase) narciclasine and siRNA that targets the unc-51-like kinase 1 (siULK1). Following this approach, a synergistic effect of inhibition of ULK1-mediated autophagy and AMPK activation resulted in effective antitumor effects in mice. The micelles were engineered with pH-responsiveness to trigger a targeted release of cargoes once they reach the acidic tumor environment [254].

To target the cell nucleus, micellar systems have been engineered to incorporate CPPs. Micelles based on chitosan-poly-(N-3-carbobenzyloxylysine) (CPCL) complexed with TAT peptide exhibited nuclear localization compared to CPCL alone. Additionally, notably enhanced cytotoxicity was observed in HeLa cells. These complexes might therefore serve as suitable nanoplatforms for the co-delivery of genes and drugs [255]. Alternatively, the amidization of the TAT peptide proved to be a promising approach to functionalize the surface of chitosan grafted carbobenzyloxylysine (CCL) for the co-delivery of the p53 gene and doxorubicin [256].

### 5.3. Liposomes

Due to their biocompatibility and ability to deliver both lipophilic and hydrophilic drugs, liposomes are considered attractive nanocarriers for gene and drug delivery systems [257]. Moreover, their internalization in cells could be enhanced by functionalizing them with CPPs. For example, liposomal delivery systems were prepared and functionalized with the antimicrobial peptide [D] H6L9 for co-delivery of anti-microRNA 10b (antagomir-10b) and paclitaxel in lung cancer cells. The peptide contains a histidine sequence and is activated at low pH, which facilitates endosomal escape. The use of microRNA is recognized as a strategy to impede metastasis, while paclitaxel simultaneously contributes to stop metastasis and promote apoptosis of tumor cells. Mice treated with these liposomes showed reduced growth of 4T1 tumors. Also, inhibition of lung metastasis was observed in vivo [258]. Following a similar approach, other studies addressed the co-delivery of paclitaxel and doxorubicin to treat melanoma by modifying liposomes with transferrin and TAT. These systems exhibited enhanced cell penetration and cytotoxic effects [259]. The functionalization of liposomes with CPPs have been proposed to create delivery systems capable of crossing the BBB. Liposomes modified with transferrin and a CPP (TAT or QLPVM) showed enhanced biodistribution across the BBB and high cytotoxicity [260].

## 6. Conclusions and Prospects

The advent of more robust gene editing platforms has opened the possibility to treat various inherited and complex diseases. Despite this enormous potential, there are still several obstacles to overcome before reaching the clinical level. These are mainly related to the specificity and assuring that enough delivered molecules reach the target tissues without compromising their functionality. In this regard, both viral and non-viral delivery vectors have been considered as suitable alternatives to tackle this main issue. In particular, non-viral vectors have attracted significant attention over the past two decades as they overcome safety issues of the viral ones. Some of the most successful vectors of this family include several types of nanomaterials and lipid-based nanocarriers. Much work has been invested to engineer their bulk and surface physicochemical properties to elicit unique interactions with multiple barriers at both the extracellular and intracellular levels. In all cases, the main goal is to come across such barriers without inducing significant perturbations that might irreversibly alter the integrity of both therapeutic cargoes and the physiological functions of the barriers themselves. This has been accomplished by synthesizing multifunctional nanovehicles that respond to carefully applied external stimuli to protect cargoes and increase specificity while intermingling with the major components of the barriers.

At the extracellular level, organs responsible for maintaining homeostasis (e.g., spleen, liver, and kidney) represent major obstacles along with complex immune responses and the presence of endothelial cell linings. In this regard, important efforts have been invested toward unraveling the mysteries of surpassing the blood-brain barrier (BBB), which has been significantly challenging due to its refractory response to most pharmacological compounds. For this reason, this remains an area of intensive research where novel vehicles should try to address the intricate cell-cell interactions to induce their temporary ease while the cargoes are transiting. In the case of intracellular obstacles, most of the research efforts have been focused on understanding and manipulating the uptake and trafficking mechanisms to reach the targeted intracellular compartments with high selectivity and efficiency. In particular, endosomal entrapment is a major challenge intracellularly, as is how cells typically deal with exogenous cargoes by attacking them with potent enzymes and very acidic environments. Several approaches have been successfully implemented to address this issue, including several surface modifications such as cationic molecules, pH-responsive materials, fusogenic agents, and compounds exhibiting buffering capacity. Here, we reviewed how these different approaches have been exploited both in vitro and in vivo, along with the perspectives for the most promising ones.

We expect that this more profound understanding of the interplay among the involved physicochemical parameters of delivery vehicles (e.g., size, morphology, surface chemistry, charge, and colloidal stability) provides clues for tailoring their next generation with higher chances of clinical translation. We are confident that future developments in this area will also be facilitated by continue working on the engineering of specific responses to external stimuli such as electrical and magnetic fields. This should also propel the development of novel bioinstrumentation to ensure that the release of cargoes is fully automated and much more controlled. Moreover, these technologies must also be adaptable to each patient’s needs and their unique biological and physiological responses.

## Figures and Tables

**Figure 1 pharmaceutics-13-00428-f001:**
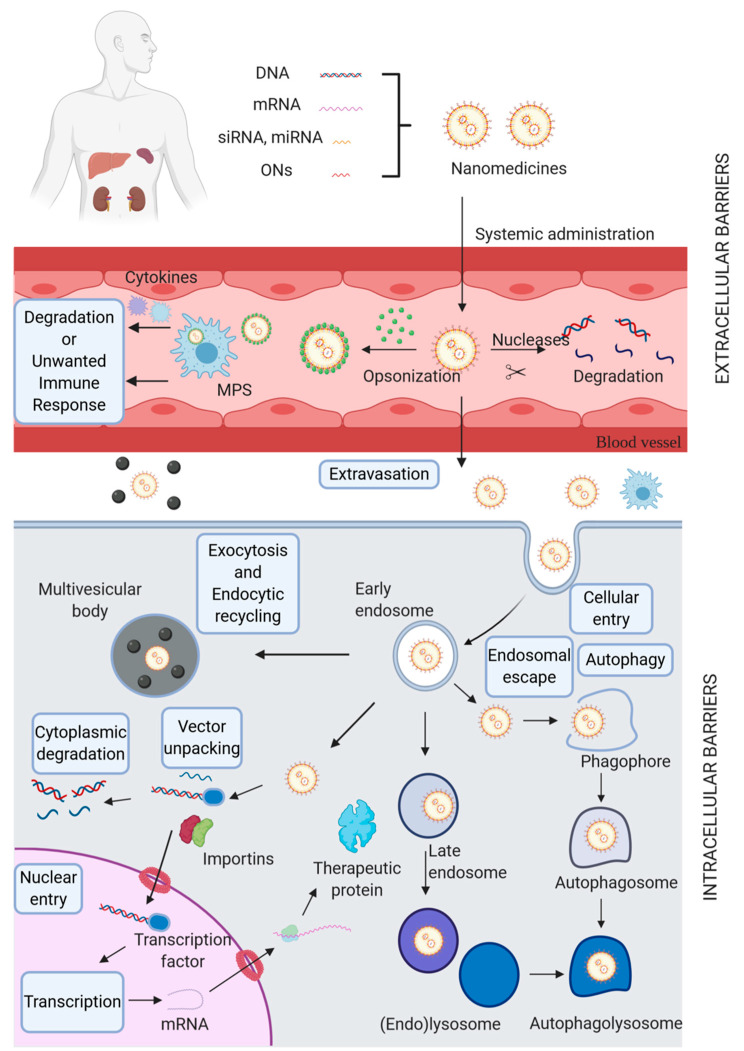
The biological barriers for non-viral gene delivery systems. Nucleic acids (e.g., DNA, mRNA, siRNA, miRNA, and oligonucleotides (ONs)) are usually incorporated into nanoparticles (NPs) to modulate protein expression levels. Extracellular (EC) barriers include: (i) the clearance through the kidney, liver, and spleen; (ii) the mononuclear phagocyte system (MPS), activated through the opsonization of the nanoparticles, with nucleases degrading the genetic cargo and finally, (iii) extravasation. Once EC barriers are circumvented, NPs must face intracellular (IC) barriers: Firstly, the plasma membrane must be crossed, usually by endocytic pathways, to ensure cellular entry. Next, NPs are encapsulated into an endosomal vesicle, which keeps them physically separated from the cytosol. Endosomal routes must be suppressed to avoid the lysosomal action responsible for degrading the internalized cargo and consequently its inability to gain access to the cytosol. Another challenge includes avoiding being recycled back to the EC environment. Once the cytosol is reached, the internalized NPs face autophagy and cytoplasmic degradation. Finally, pDNA must overcome the nuclear envelope to be delivered into the nucleus. Figure based on the work of Vermeulen et al. and Okholm et al. [12,13]. Created with BioRender.com.

**Figure 2 pharmaceutics-13-00428-f002:**
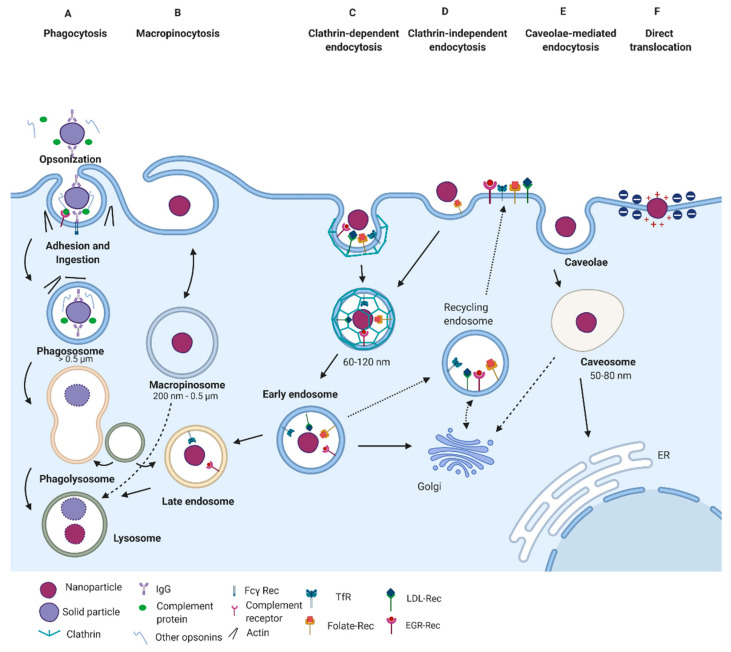
Mechanisms of internalization in living cells. (**A**) Phagocytosis; (**B**) Macropinocytosis; (**C**) Clathrin-dependent endocytosis; (**D**) Clathrin-independent endocytosis; (**E**) Caveolae-mediated endocytosis; (**F**) Direct translocation. Other conventions: IgG, Immunoglobulin G; Fcγ Rec, Fcγ receptor; TfR, Transferrin receptor; Folate-Rec, Folate receptor; LDL-Rec, low-density lipoprotein receptor; EGF-Rec, Epidermal growth factor receptor; ER, Endoplasmatic reticulum. This figure was based on Yameen et al. and Hillaireau et al. [48,49]. Created with BioRender.com.

**Figure 3 pharmaceutics-13-00428-f003:**
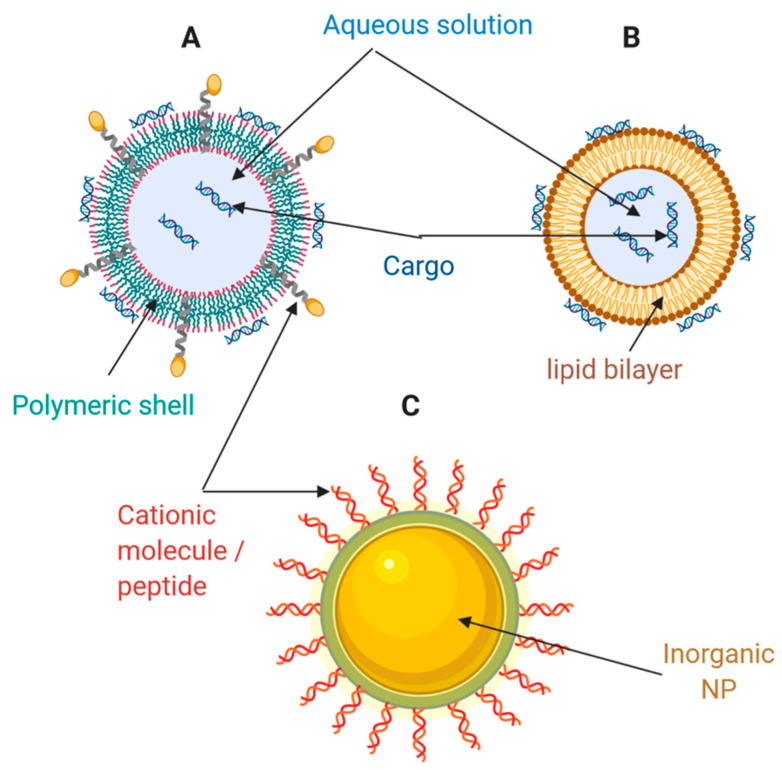
Schematic of different types of nanocarriers for drug and gene delivery. (**A**) Polymeric nanoparticle; (**B**) Liposome; (**C**) Inorganic nanoparticle. Adapted from Weng et al. [90], Molecular Therapy Nucleic Acids, 2020.

**Figure 4 pharmaceutics-13-00428-f004:**
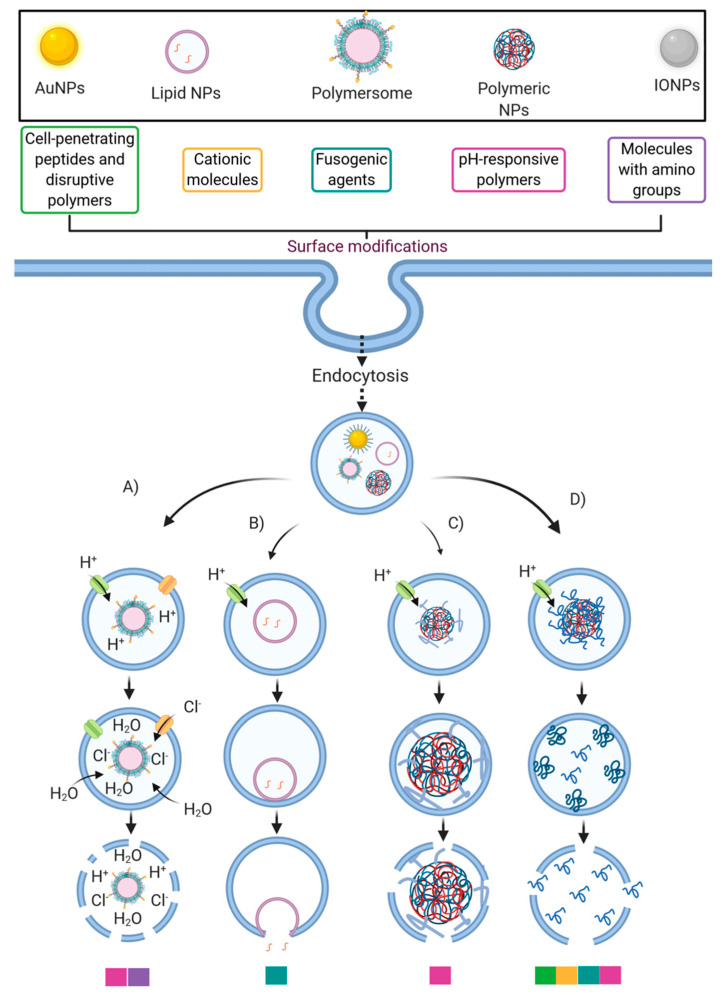
Strategies to improve the endosomal escape ability of some nanocarriers. This schematic illustrates the association between the more typically employed surface modifications and the endosomal escape mechanisms (EEMs): (**A**) Proton-Sponge and osmotic lysis; (**B**) Membrane fusion; (**C**) Particle swelling, and (**D**) Membrane translocation and destabilization. The color legend below the EEMs corresponds to the surface modifications that likely trigger the corresponding EEM. Created with BioRender.com.

**Figure 5 pharmaceutics-13-00428-f005:**
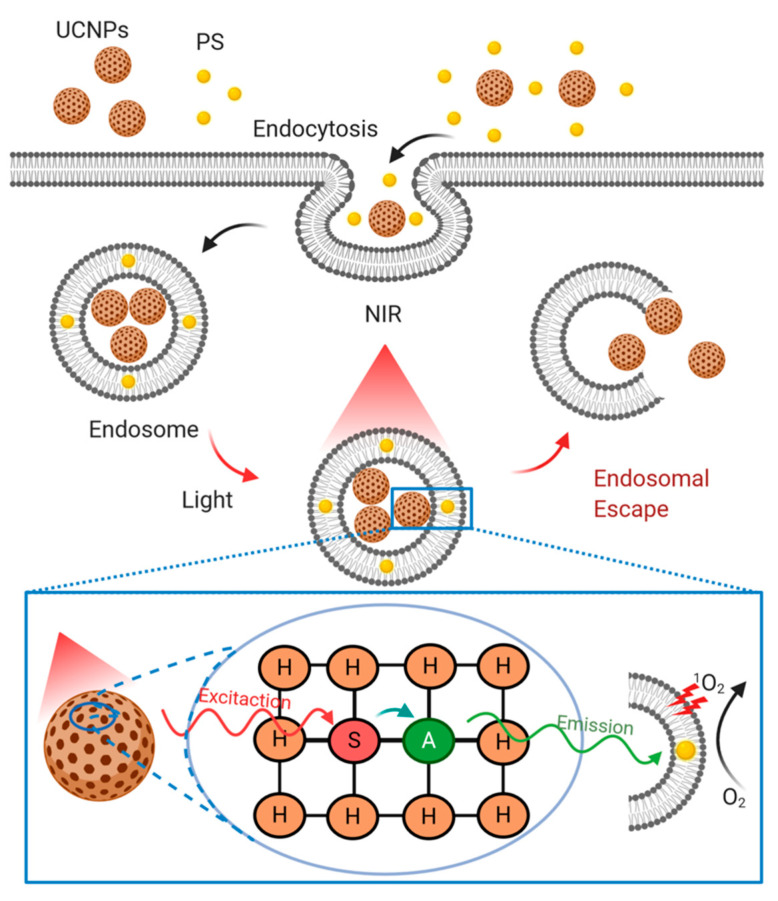
Schematic of the photochemical internalization followed by UCNPs to escape from endosomal entrapment. UCNPs and photosensitizers (PS) are taken up by the cell via endocytosis and co-localized with endosomes. PS intercalate within endosomal membranes due to their amphiphilic properties. After NIR irradiation, the energy absorbed by sensitizer ions (S) is transferred to activator ions (A), then emitting radiation, either in the UV or Vis range. Next, the PS absorbs the activator ion’s energy and transfers it to molecular oxygen to produce highly toxic singlet oxygen, which causes severe damage to the endosomal membrane due to its oxidative effects on amino acids (e.g., tryptophan, cysteine, histidine, methionine, and phenylalanine), unsaturated fatty acids and cholesterol. Ultimately, membrane disruption is achieved, and UCNPs escape from endosomes. H: host matrix. This schematic was based on Rueda-Gensini et al. [73]. Created with BioRender.com.

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
