# Peer review of "Delivery Systems for Nucleic Acids and Proteins: Barriers, Cell Capture Pathways and Nanocarriers"

_pharmaceutics, 2021, doi:10.3390/pharmaceutics13030428_

Round 1
Reviewer 1 Report
The manuscript "Delivery systems for nucleic acids and proteins: Barriers, cell capture pathways and nanocarriers" is a well written and organized review about delivery systems. I suggest to publish it without further changes.
The authors focus their attention on non-viral delivery systems and deeply describe either cell internalization mechanisms and bypass biological barriers.
They describe different types of nanocarriers from lipid-based to polymeric and finally inorganic nanomaterials with an appropriate and up-to-date number of references.
Author Response
The manuscript "Delivery systems for nucleic acids and proteins: Barriers, cell capture pathways and nanocarriers" is a well written and organized review about delivery systems. I suggest to publish it without further changes.
The authors focus their attention on non-viral delivery systems and deeply describe either cell internalization mechanisms and bypass biological barriers.
They describe different types of nanocarriers from lipid-based to polymeric and finally inorganic nanomaterials with an appropriate and up-to-date number of references.
Response: We are very thankful to the reviewer for the positive evaluation of our contribution
Reviewer 2 Report
The review under evaluation deals with an interesting topic and is illustrated with very nice figures. The text is well organised, but some sections present some issues in terms of clarity. Here below are presented some suggested change and inputs for the improvement of the manuscript.
Line 48 - I suggest to use as few lines before "chemical constructed vectors" and not "chemical conjugation-based vectors" since chemical conjugation implies a covalent bond (electrostatic interaction is not a conjugation but complexation).
Lines 57-59. The different strategy in the use of DNA or RNAi based therapies (and maybe also the others introduced in Figure 1) should be explained in more detail here, as it is not introduced elsewhere.
Line 74. I would use "extensive" or "comprehensive", not both together.
Line 95. I suggest to omit the word "circulation".
Lines 94-101. The phrase starting from "Also..." and ending with "...space [12]" is lacking of the main verb. Also the following one presents some issues... It starts mentioning as topical administration oral and pulmonary delivery and ends stating that this approach increase the targeting to ocular areas (!). Obviously this latter statement is true for local ocular administration, but again the construction of the phrase is far from optimal. Please rewrite the section.
Line 166 - It is not clear what is the "large excess of dsDNA" and in any case the rationale of the experiment is not evident.
Line 199 - Generally, "Ran" is indicating the protein while "RAN" is the expressing it. Please correct.
Line 360 - It is not clear if "including Arf-6, flotillin, Cdc-42, and RhoA" is referred to the "effectors" mentioned before, for sure are not the cargoes. Please rewrite to make more clear.
Line 363 - it should be "Rab5-positive".
Line 404 - "Lipid emulsions" is a misnomer, since emulsions always contain an oil phase. The only differentiation could be between O/W or W/O emulsions. It is suggested to use only the term "emulsions".
Line 442 - It is not clear what a "bio-orthogonal group" is. Please explain.
Line 481 - "serving as serve as".
Line 497 - It is not clear what is meant with the attribute "radiation-like". Please eliminate the word.
Lines 560-1 - In the sentence "Na et al. [154] demonstrated efficient target gene knockdown, in vitro and in vivo, for MSN with similar pore size and negligible cytotoxicity." is not clear what is the target gene, nor the pore size (similar to what?) or the type of cells used for toxicity. Please avoid here and elsewhere in the manuscript, referring to papers in a way that do not provide any information to the reader.
Lines 697-717 - UCNPs use for ibuprofen or doxorubicin delivery appears out of the scope of the review and should be eliminated.
Line 929 - on the toxicity of CNT on airways epithelia see also Nanotoxicology 29 (2015) 1-12 (doi:10.3109/17435390.2014.918203)
Please carefully control the text for typos and grammatical errors.
Author Response
The review under evaluation deals with an interesting topic and is illustrated with very nice figures. The text is well organised, but some sections present some issues in terms of clarity. Here below are presented some suggested change and inputs for the improvement of the manuscript.
- Line 48 - I suggest to use as few lines before "chemical constructed vectors" and not "chemical conjugation-based vectors" since chemical conjugation implies a covalent bond (electrostatic interaction is not a conjugation but complexation).
Response: According to the reviewer's suggestion, the term chemical conjugation was replaced by chemically-constructed.
- Lines 57-59. The different strategy in the use of DNA or RNAi based therapies (and maybe also the others introduced in Figure 1) should be explained in more detail here, as it is not introduced elsewhere.
Response: We agree with the reviewer that the DNA and RNAi therapies should have been introduced earlier in the discussion. Accordingly, more details on such discussion were added to the revised manuscript. Also, Figure 1 was altered to incorporate missing details.
- Line 74. I would use "extensive" or "comprehensive", not both together.
Response: According to the reviewer’s suggestion, we picked comprehensive over extensive for our argument.
- Line 95. I suggest to omit the word "circulation".
Response: According to the reviewer’s suggestion, we omitted the word circulation.
- Lines 94-101. The phrase starting from "Also..." and ending with "...space [12]" is lacking of the main verb. Also the following one presents some issues... It starts mentioning as topical administration oral and pulmonary delivery and ends stating that this approach increase the targeting to ocular areas (!). Obviously this latter statement is true for local ocular administration, but again the construction of the phrase is far from optimal. Please rewrite the section.
Response: We are very grateful to the reviewer for pointing this inconsistency out. We therefore rewrote the whole section for clarity.
- Line 166 - It is not clear what is the "large excess of dsDNA" and in any case the rationale of the experiment is not evident.
Response: According to the reviewer’s suggestion, we explained the rationale behind the discussed experiment and what is meant by large excess of dsDNA.
- Line 199 - Generally, "Ran" is indicating the protein while "RAN" is the expressing it. Please correct.
Response: The name was corrected according to the reviewer’s suggestion.
- Line 360 - It is not clear if "including Arf-6, flotillin, Cdc-42, and RhoA" is referred to the "effectors" mentioned before, for sure are not the cargoes. Please rewrite to make more clear.
Response: In line with the reviewer’s suggestion, we clarified how Arf-6, flotillin, Cdc-42, and Rho are involved in the discussion.
- Line 363 - it should be "Rab5-positive".
Response: The suggestion by the reviewer was included in revised text.
- Line 404 - "Lipid emulsions" is a misnomer, since emulsions always contain an oil phase. The only differentiation could be between O/W or W/O emulsions. It is suggested to use only the term "emulsions".
Response: We absolutely agree with the reviewer and corrected the misnomer to only make a reference to emulsions.
- Line 442 - It is not clear what a "bio-orthogonal group" is. Please explain.
Response: The concept of bio-orthogonal group was clarified in the revised version of the manuscript.
- Line 481 - "serving as serve as".
Response: The sentence was adjusted according to the reviewer’s suggestion
- Line 497 - It is not clear what is meant with the attribute "radiation-like". Please eliminate the word.
Response: The “radiation-like” attribute was removed from the text
- Lines 560-1 - In the sentence "Na et al. [154] demonstrated efficient target gene knockdown, in vitro and in vivo, for MSN with similar pore size and negligible cytotoxicity." is not clear what is the target gene, nor the pore size (similar to what?) or the type of cells used for toxicity. Please avoid here and elsewhere in the manuscript, referring to papers in a way that do not provide any information to the reader.
Response: The sentence was rewritten to make sure that the stated comparison is clear in terms of the involved genes and the attibutes of the material. Also, we removed most of the ambiguous references to previous contributions.
- Lines 697-717 - UCNPs use for ibuprofen or doxorubicin delivery appears out of the scope of the review and should be eliminated.
Response: We removed the discussion on the delivery of small pharmacological molecules aided by UCNPs.
- Line 929 - on the toxicity of CNT on airways epithelia see also Nanotoxicology 29 (2015) 1-12 (doi:10.3109/17435390.2014.918203)
Response: We included the suggested reference in the revised manuscript.
- Please carefully control the text for typos and grammatical errors.
Response: The manuscript was carefully reviewed to remove typos and grammar mistakes.
Reviewer 3 Report
In the review by Julian Daniel Torres et al., authors investigate different nanovehicles for the delivery of nucleic acids and proteins. The research topic is appealing and actual, it explains several interesting points; figures and tables are explicative and overall is well written. In my opinion authors should remove the word “proteins” from the title because the discussion is minimal compared to the focus on the nucleic acids; otherwise, the title could be also changed in something more fascinating. Nevertheless, as I mentioned before the work is well written and thus, I would suggest being published as it is.
Author Response
In the review by Julian Daniel Torres et al., authors investigate different nanovehicles for the delivery of nucleic acids and proteins. The research topic is appealing and actual, it explains several interesting points; figures and tables are explicative and overall is well written. In my opinion authors should remove the word “proteins” from the title because the discussion is minimal compared to the focus on the nucleic acids; otherwise, the title could be also changed in something more fascinating. Nevertheless, as I mentioned before the work is well written and thus, I would suggest being published as it is.
Response: We are very thankful to the reviewer for the positive evaluation of our contribution.
Reviewer 4 Report
This review is an interesting paper. It makes a useful contribution to the growing literature on successful strategies to deliver DNA and proteins. The literature review is thorough, with important points, well referenced and very well analyzed. It should be of great interest to the readers.
I have quite a few, more minor comments, and these are listed below.
- The manuscript is in need of proof reading for English by a native speaker.
- Lines 93-94: “EC barriers need to be overcome before reaching the target cell. These include organs such as the spleen, liver, and kidney that uptake or filter nanoparticles”· These sentences show theses organs as a barriers but maybe they are confusing. I suggest explaining better why they are included as a barrier and how the nanoparticle is eliminated.
- 3.1 Phagocytosis. Authors should be provided an evidence of microglia in the nervous system.
- Figure 2. In this figure lacks some abbreviations.
Author Response
This review is an interesting paper. It makes a useful contribution to the growing literature on successful strategies to deliver DNA and proteins. The literature review is thorough, with important points, well referenced and very well analyzed. It should be of great interest to the readers.
I have quite a few, more minor comments, and these are listed below.
- The manuscript is in need of proof reading for English by a native speaker.
Response: The entire manuscript was proof-read by a native speaker of the English language.
- Lines 93-94: “EC barriers need to be overcome before reaching the target cell. These include organs such as the spleen, liver, and kidney that uptake or filter nanoparticles”· These sentences show theses organs as a barriers but maybe they are confusing. I suggest explaining better why they are included as a barrier and how the nanoparticle is eliminated.
Response: According to the reviewer suggestion, we expanded on the role of EC barriers in organ delivery and on the fate of nanoparticles once they complete their task.
- 3.1 Phagocytosis. Authors should be provided an evidence of microglia in the nervous system.
Response: We included additional evidence of phagocytosis in microglia.
- Figure 2. In this figure lacks some abbreviations.
Response: We are very grateful to the reviewer for pointing this out. The missing abbreviations were included accordingly.
Reviewer 5 Report
The manuscript entitled “Delivery systems for nucleic acids and proteins: Barriers, cell capture pathways and nano-carriers” submitted by Torres and collaborators reviews the delivery systems for nucleic acids and proteins and highlights the different obstacles (barriers) and how to overcome these limitations by using diverse nano-carriers. This review provides an interesting survey of the different delivery systems and their limitations and propose some alternatives to improve the delivery systems
This review is well constructed and gives a large overview of the actual knowledge of delivery systems. However, it is too long and sometimes it is hard to read. Before accepting this review for its publication in Pharmaceutics journal, the authors need to make corrections cited below.
Some paragraphs are too much developed or are out of focus of this review:
- The Blood-Brain-Barrier (BBB°: the description of the different cell types of the BBB is not necessary for this review.
- The “Internalization pathways” is also too long. Different reviews have already described this pathway. Indeed in this part of the review, the direct translocation is missing in this part and in figure 2.
The authors review different nano-carriers but they don’t review the CPP (Cell Penetrating Peptides) as nano-carriers by themselves. During the past two decades, there a plenty works devoted to CPPs as nano-vectors
Author Response
The manuscript entitled “Delivery systems for nucleic acids and proteins: Barriers, cell capture pathways and nano-carriers” submitted by Torres and collaborators reviews the delivery systems for nucleic acids and proteins and highlights the different obstacles (barriers) and how to overcome these limitations by using diverse nano-carriers. This review provides an interesting survey of the different delivery systems and their limitations and propose some alternatives to improve the delivery systems
This review is well constructed and gives a large overview of the actual knowledge of delivery systems. However, it is too long and sometimes it is hard to read. Before accepting this review for its publication in Pharmaceutics journal, the authors need to make corrections cited below.
Some paragraphs are too much developed or are out of focus of this review:
- The Blood-Brain-Barrier (BBB°: the description of the different cell types of the BBB is not necessary for this review.
Response: By following the reviewer’s recommendation, we removed the details on the types of cells forming the BBB.
- The “Internalization pathways” is also too long. Different reviews have already described this pathway. Indeed in this part of the review, the direct translocation is missing in this part and in figure 2.
Response: The section referring to internalization pathways was shortened and we included the missing information regarding direct translocation.
- The authors review different nano-carriers but they don’t review the CPP (Cell Penetrating Peptides) as nano-carriers by themselves. During the past two decades, there a plenty works devoted to CPPs as nano-vectors
Response: We are very thankful to the reviewer for encouraging us to make a more robust description of all possible nanovehicles for delivery. In this regard, we included a discussion on CPPs as emerging nanocarriers.
Round 2
Reviewer 5 Report
The authors improved their manuscript which is now acceptable for publication in Pharmaceutics journal